# Separate Anything in Audio with Zero Training

**Chao Huang**[1], **Yuesheng Ma**[2], **Junxuan Huang**[3], **Susan Liang**[1], **Yunlong Tang**[1],
**Jing Bi**[1], **Wenqiang Liu**[3], **Nima Mesgarani**[2], **Chenliang Xu**[1]
[1]University of Rochester, [2]Columbia University, [3]Tencent America

## Abstract

Audio source separation is fundamental for machines to understand complex acoustic environments and underpins numerous audio applications. Current supervised deep learning approaches, while powerful, are limited by the need for extensive, task-specific labeled data and struggle to generalize to the immense variability and open-set nature of real-world acoustic scenes. Inspired by the success of generative foundation models, we investigate whether pre-trained text-guided audio diffusion models can overcome these limitations. We make a surprising discovery: zero-shot source separation can be achieved purely through a pre-trained text-guided audio diffusion model under the right configuration. Our method, named `ZeroSep`, works by inverting the mixed audio into the diffusion model's latent space and then using text conditioning to guide the denoising process to recover individual sources. Without any task-specific training or fine-tuning, `ZeroSep` repurposes the generative diffusion model for a discriminative separation task and inherently supports open-set scenarios through its rich textual priors. `ZeroSep` is compatible with a variety of pre-trained text-guided audio diffusion backbones and delivers strong separation performance on multiple separation benchmarks, surpassing even supervised methods. Our project page is here: `https://wikichao.github.io/ZeroSep/`.

## 1 Introduction

At the heart of acoustic scene perception lies the fundamental task of source separation, which aims to isolate individual sound sources from a complex audio mixture. Accurate source separation is crucial for a wide range of applications, including media production, surveillance systems, automatic speech recognition in noisy environments, and analysis of complex soundscapes.

The dominant approach to audio separation in recent years has relied heavily on supervised learning: deep neural networks are trained on large datasets of paired mixtures and clean sources [Luo and Mesgarani, 2019, Subakan et al., 2021]. While these methods have achieved impressive performance on specific, well-represented source types and datasets, they often fall short when faced with the open-set variability of real-world acoustic scenes. Consequently, training a foundation-level separation model becomes exceptionally challenging due to the need for vast amounts of labeled data, the difficulty of defining training objectives and mixing strategies, and the design of effective conditioning mechanisms.

Recent efforts, such as LASS-net Liu et al. [2022a], AudioSep Liu et al. [2023a], and FlowSep Yin et al. [2024], have explored leveraging natural language queries for more flexible separation. Despite these advances, they still contend with the same core challenges: vast data requirements, complex task-specific training regimes, and limited generalization to unseen acoustic scenes. Inspired by the transformative success of large language models in unifying diverse NLP tasks under a generative framework [Brown et al., 2020], we pose a central question: *Can a generative foundation model similarly emerge for audio tasks?* In this work, we explore this question by investigating the capabilities of pre-trained text-guided audio diffusion models.

39th Conference on Neural Information Processing Systems (NeurIPS 2025).

We discover that a text-guided audio diffusion model can, *out of the box*, separate a mixture into its sources – **no training or fine-tuning**, relying solely on latent inversion and conditioned denoising: (i) Given a mixed audio signal, we can find a corresponding point in the diffusion model's latent space through an inversion process. This latent representation captures the composite information from all the sound sources present in the mixture. (ii) Subsequently, by guiding the generative denoising process from this latent state using text prompts corresponding to individual sources in the mixture, the model can be steered to reconstruct each source in isolation. Surprisingly, even though this is a generative process, the separated sources are highly faithful to the original sources, especially with classifier-free guidance $\leq 1$, which prevents hallucination. This effectively repurposes the generative model for a discriminative task, offering a fundamentally different approach to separation.

Based on the above observations, we introduce `ZeroSep`, a zero-training framework for audio source separation that repurposes pretrained text-guided diffusion models. By casting separation as a two-step generative inference, latent inversion followed by text-conditioned denoising, `ZeroSep` offers three key advantages:

*Open-set Separation:* As the core of `ZeroSep` is a pre-trained text-guided audio diffusion model, which has learned to generate realistic audios from diverse, open-domain descriptions and mixing styles, `ZeroSep` naturally handles open-set queries and is able to separate from diverse mixtures.

*Model-agnostic Versatility:* The inversion plus denoising pipeline is generic to diffusion architectures, allowing `ZeroSep` to leverage different pre-trained audio diffusion backbones. Interestingly, we observe a trend that the better the audio diffusion model can generate, the better it can separate, which could suggest continuous improvement whenever there is a more advanced audio generation model available.

*Training-Free Efficacy:* Without any fine-tuning or task-specific data, `ZeroSep` matches or exceeds the performance of existing training-based generative separators, overturning the assumption that high-quality separation requires dedicated training.

In summary, our contributions to the community includes

1. We introduce `ZeroSep`, a *training-free* audio source-separation framework that repurposes pre-trained text-guided diffusion models, representing a fundamental shift away from supervised separation paradigms.

2. We demonstrate that pure generative inference—latent inversion followed by text-conditioned denoising—yields state-of-the-art separation performance, outperforming existing training-based generative methods.

3. We establish `ZeroSep` 's versatility and open-set capability: it seamlessly handles diverse mixtures and textual queries and can be applied to many pre-trained audio diffusion backbones, improving separation quality as the underlying model's generative fidelity increases.

## 2 Related Works

**Audio Diffusion Models.** Diffusion probabilistic models have rapidly emerged as a leading paradigm for generating high-quality and diverse audio content. Early works like DiffWave [Kong et al., 2021] and WaveGrad [Chen et al., 2021a] demonstrated the potential of applying denoising diffusion to synthesize raw audio waveforms, achieving high-fidelity unconditional audio generation. Building on this foundation, diffusion models were successfully extended to conditional audio generation tasks. In text-to-speech (TTS), models such as Diff-TTS [Jeong et al., 2021] and Grad-TTS [Popov et al., 2021] showed that diffusion processes could generate high-fidelity mel-spectrograms conditioned on text input. Researchers also focused on improving the efficiency and controllability of diffusion sampling; for instance, Guided-TTS Kim et al. [2022] introduced classifier guidance for TTS, and PriorGrad [Lee et al., 2022] addressed sampling speed in vocoders through data-dependent priors. Diffusion models have also been applied to other audio synthesis tasks, including singing voice synthesis with DiffSinger [Liu et al., 2022b] and waveform super-resolution with NU-Wave [Lee and Han, 2021]. More recently, the focus has shifted towards latent-space diffusion models and text-conditioned generation of general audio. AudioLDM [Liu et al., 2023b] pioneered combining diffusion with CLAP embeddings to enable text-conditioned generation of diverse sounds and music. AudioLDM2 [Liu et al., 2024] and Tango [Ghosal et al., 2023] further advanced in this direction, providing enhanced control and quality. These text-conditioned latent diffusion models [Evans et al.,

2025a], capable of generating complex audio scenes from natural language, form the technological foundation for our training-free separation method `ZeroSep`.

**Audio Separation.** The problem of source separation has long been tackled by both classic signal-processing techniques and, more recently, deep learning. Traditional methods such as NMF-MFCC [Stöter et al., 2021] decompose mixtures under assumptions about timbral or spectral structure. While training-free, they often fail on complex or heavily overlapping sources that lack clear distinguishing features. Deep learning revolutionized the field by learning representations directly from data. Deep Clustering [Hershey et al., 2016] trains embeddings for clustering source-specific time–frequency bins, and Permutation-Invariant Training (PIT) [Yu et al., 2017] resolves the label-permutation problem during training. Conv-TasNet [Luo and Mesgarani, 2019] further advanced performance with end-to-end waveform separation, frequently surpassing traditional masking approaches. Diffusion-based approaches such as SepDiff [Chen et al., 2023] and DiffSep [Scheibler et al., 2023] reformulate separations as a generative problem. However, these models remain "blind" to user intent: once trained, they separate every detectable component rather than targeting a specific source. To introduce controllability, recent works condition separation on auxiliary modalities. Video-guided methods [Huang et al., 2024a] use visual cues, while language-based frameworks, such as LASS-Net [Liu et al., 2022a], AudioSep [Liu et al., 2023a], and FlowSep [Yin et al., 2024], leverage text prompts to guide mask estimation. Although more flexible, they still require large supervised corpora of synthetic mixtures, inheriting closed-world biases. Zero-shot diffusion editors like AUDIT [Wang et al., 2023], DITTO [Novack et al., 2024], and AudioEdit [Manor and Michaeli, 2024] fine-tune or invert latent trajectories to delete components, but focus on editing rather than explicit separation.

In contrast, `ZeroSep` repurposes a pre-trained text-guided audio diffusion model as a universal, training-free [Postolache et al., 2024] prior for open-set separation. By (i) inverting an audio mixture into the model's latent space and (ii) re-denoising under user-provided text prompts with unit classifier-free guidance, `ZeroSep` generates one isolated waveform per prompt, achieving comprehensive, zero-shot source separation without fine-tuning.

## 3 Method

In this section, we first review the foundational knowledge of text-guided diffusion models and diffusion inversion techniques, which form the basis of our method. Next, we discuss the separation task setup with generative diffusion models. Lastly, we introduce `ZeroSep`, a zero-shot separation adaptation of existing text-guided audio diffusion models.

### 3.1 Preliminary: Text-Guided Audio Diffusion and Inversion

Text-guided audio diffusion models typically operate in a learned latent space: An initial audio signal is first encoded into a latent representation, denoted as $x_0$; the forward diffusion process progressively adds Gaussian noise to this latent vector, transforming it into a pure noise vector $x_T$. A neural network, parameterized by a set of $\theta$, learns to predict and remove the noise added at each step $t$, effectively reversing the diffusion process and generating mel-spectrograms which are then converted to waveforms using a vocoder[1]. In text-guided models, this denoising process is driven by a text condition $c$, derived from a text encoder, ensuring the generated audio aligns with the text prompt.

**DDIM Inversion.** To enable manipulation of existing audio content, inversion techniques are used to map a real audio sample back into the noisy latent space. A common approach is DDIM inversion, which leverages the deterministic nature of DDIM sampling [Song et al., 2020]. The standard DDIM sampling process iteratively denoises a noisy latent $\mathbf{x}_t$ to produce a less noisy version $\mathbf{x}_{t-1}$:

$$\mathbf{x}_{t-1} = \sqrt{\tfrac{\bar{\alpha}_{t-1}}{\bar{\alpha}_t}}\, \mathbf{x}_t + \left( \sqrt{\tfrac{1}{\bar{\alpha}_{t-1}} - 1} - \sqrt{\tfrac{1}{\bar{\alpha}_t} - 1} \right) \epsilon_\theta\big(\mathbf{x}_t, \mathbf{c}, t\big), \tag{1}$$

---

[1]This pipeline underlies many audio diffusion models, such as the AudioLDM family [Liu et al., 2024, 2023b] and Tango [Ghosal et al., 2023], but does not apply to architectures like Stable Audio Open [Evans et al., 2025a].

where $\{\bar{\alpha}_t\}_{t=0}^T$ defines the noise schedule and $\epsilon_\theta(\cdot, \mathbf{c}, t)$ is the model's noise prediction conditioned on $\mathbf{c}$. DDIM inversion reverses this process, estimating the noisy latent $\mathbf{x}_{t+1}$ from $\mathbf{x}_t$:

$$\mathbf{x}_{t+1} = \sqrt{\frac{\bar{\alpha}_{t+1}}{\bar{\alpha}_t}} \, \mathbf{x}_t + \left( \sqrt{\frac{1}{\bar{\alpha}_{t+1}} - 1} - \sqrt{\frac{1}{\bar{\alpha}_t} - 1} \right) \epsilon_\theta(\mathbf{x}_t, \mathbf{c}, t), \tag{2}$$

so that iterating from $\mathbf{x}_0$ recovers an estimate of the pure noise $\mathbf{x}_T$. Cumulative errors can, however, cause deviations from the true noise trajectory.

**DDPM Inversion.** In contrast to DDIM inversion, DDPM inversion [Huberman-Spiegelglas et al., 2024] leverages the probabilistic forward diffusion to obtain an exact noise path. Given a clean latent $\mathbf{x}_0$, one constructs an auxiliary sequence of noisy latents

$$\mathbf{x}_t = \sqrt{\bar{\alpha}_t} \, \mathbf{x}_0 + \sqrt{1 - \bar{\alpha}_t} \, \tilde{\epsilon}_t, \quad \tilde{\epsilon}_t \sim \mathcal{N}(\mathbf{0}, \mathbf{I}), \; t = 1, \dots, T, \tag{3}$$

and then extracts the per-step noise vectors

$$\mathbf{z}_t = \frac{\mathbf{x}_{t-1} - \mu_t(\mathbf{x}_t)}{\sigma_t}, \quad t = T, \dots, 1, \tag{4}$$

where $\mu_t(\mathbf{x}_t)$ and $\sigma_t$ follow the DDPM [Ho et al., 2020] reverse-step definitions. Reconstruction simply re-injects $\mathbf{x}_T$ and $\{\mathbf{z}_t\}$ via

$$\mathbf{x}_{t-1} = \mu_t(\mathbf{x}_t) + \sigma_t \mathbf{z}_t, \tag{5}$$

exactly recovering $\mathbf{x}_0$. By scaling or replacing $\{\mathbf{z}_t\}$ (e.g. using text embeddings $\mathbf{c}$ at select timesteps), DDPM inversion offers a probabilistic framework for precise, text-guided edits.

From a high-level perspective, both DDIM and DDPM inversion can be viewed as a single mapping, which we denote by $\mathbf{x}_T = \mathbf{F}^{\text{inv}}(\mathbf{x}_0, \mathbf{c})$. This operator $\mathbf{F}^{\text{inv}}$ encapsulates the step-wise recovery of the noise trajectory corresponding to a given clean latent. Whether implemented via the deterministic updates of DDIM or the probabilistic steps of DDPM, $\mathbf{F}^{\text{inv}}$ produces the pure noise $\mathbf{x}_T$ that, when re-injected into the standard diffusion sampler, exactly reconstructs the original sample $\mathbf{x}_0$.

### 3.2 Task Setup

In real-world scenarios, an audio stream $a$ can be a mixture of $N$ individual sound sources: $a = \sum_{i=1}^N s^{(i)}$, where each source $s^{(i)}$ can be of various categories. To work in the diffusion latent space, we first convert $a$ to a mel-spectrogram and encode it with a Variational Autoencoder (VAE), yielding latent features $\mathbf{x} \in \mathbb{R}^{C \times T \times F}$, where $C$ is the number of channels, $F$ the numbers of frequency bins, and $T$ the number of time frames. Let $\mathbf{x}^{\text{mix}}$ denote the VAE encoding of the mixture and $\mathbf{x}^{(i)}$ the encoding of source $i$. Our goal is to find a separation mapping $f(\mathbf{x}^{\text{mix}}, \mathbf{c}^{(i)}) \longrightarrow \mathbf{x}^{(i)}$, where $\mathbf{c}^{(i)}$ is a conditioning signal (e.g., text description) that specifies which source to extract. $\mathbf{x}^{(i)}$ is then fed to the VAE decoder and Vocoder to convert latent features back to waveform level to obtain $\hat{s}^{(i)}$.

### 3.3 From Generation to Separation: The `ZeroSep` Principle

The core of `ZeroSep` lies in repurposing a pre-trained text-guided audio diffusion model, originally designed for generating audio from text, to perform the task of audio source separation.

Let $\mathbf{c}_{\text{inv}}$ be the text prompt used during the inversion process (mapping the mixed audio $\mathbf{x}^{\text{mix}}$ to a noisy latent $\mathbf{x}_T$), and $\mathbf{c}_{\text{rev}}$ be the text prompt used during the subsequent reverse denoising process (reconstructing a clean source from the noisy latent).

Diffusion models typically employ classifier-free guidance during denoising, where the noise prediction $\epsilon_t$ at step $t$ is a combination of an unconditional prediction and a conditional prediction guided by $\mathbf{c}_{\text{rev}}$:

$$\epsilon_t = \epsilon_\theta(\mathbf{x_t}, \varnothing, t) + \omega \cdot (\epsilon_\theta(\mathbf{x_t}, \mathbf{c}_{\text{rev}}, t) - \epsilon_\theta(\mathbf{x_t}, \varnothing, t)). \tag{6}$$

Here, $\epsilon_\theta(\mathbf{x_t}, \varnothing, t)$ is the unconditional noise prediction, $\epsilon_\theta(\mathbf{x_t}, \mathbf{c}_{\text{rev}}, t)$ is the prediction guided by $\mathbf{c}_{\text{rev}}$, and $\omega$ is the classifier-free guidance weight controlling the influence of the text condition $\mathbf{c}_{\text{rev}}$.

While this formulation is typically used to amplify the presence of the desired content during generation, we discover that specific choices of $\mathbf{c}_{\text{inv}}$, $\mathbf{c}_{\text{rev}}$, and $\omega$, enable effective source separation. This shifts the model's function from synthesizing new audio to dissecting existing mixtures. Here are the key principles for transforming the generative process into a separation tool:

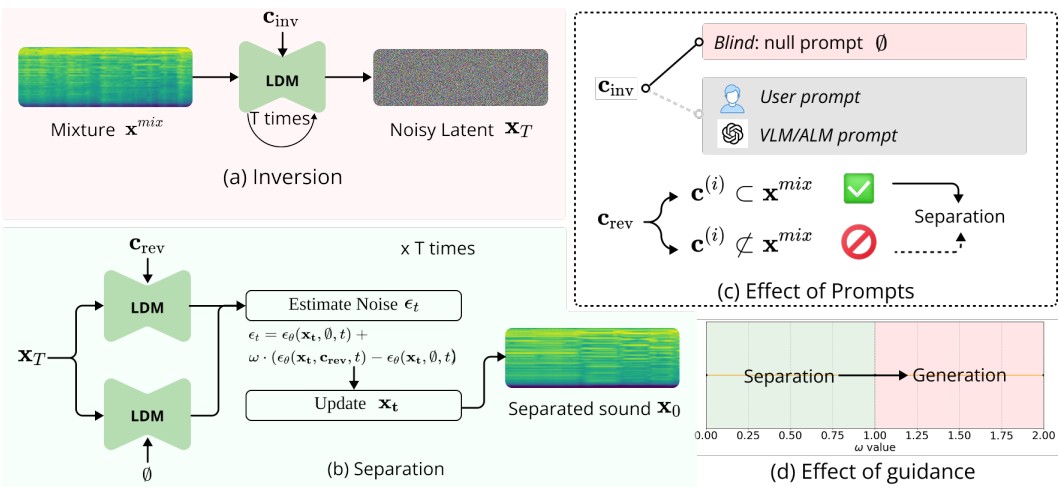

Figure 1: **The overview of** `ZeroSep`, which includes (a) an inversion process to obtain a latent representation for the mixture, and (b) a separation denoising process to effectively extract the target source with text conditions. We show the choice of inversion prompt $\mathbf{c}_{\text{inv}}$ and reverse prompt $\mathbf{c}_{\text{rev}}$ in (c), and demonstrate the valid separation region defined by $\omega$ in (d).

– **The Reverse Prompt $\mathbf{c}_{\text{rev}}$:** To isolate a specific source $i$, the reverse denoising prompt $\mathbf{c}_{\text{rev}}$ must explicitly describe that target source:

$$\mathbf{c}_{\text{rev}} := \mathbf{c}^{(i)}, \quad \text{if separating source } i. \tag{7}$$

This directs the denoising process to reconstruct the audio components associated with the target source described by $\mathbf{c}^{(i)}$. Using any other prompt would result in guided generation, not separation.

– **The Inversion Prompt $\mathbf{c}_{\text{inv}}$:** The inversion prompt $\mathbf{c}_{\text{inv}}$ influences how the mixed audio is mapped to the noisy latent space. We found flexibility here, with effective choices including a null prompt $\varnothing$ or prompts describing the other sources present in the mixture ($\mathbf{c}^{(j)}$ for $j \neq i$). While describing other sources can potentially refine the latent representation by emphasizing non-target components, it requires prior knowledge of the mixture's contents, yet can be achieved with user query or by prompting Vision-Language Models or Audio Language Models (as shown in Fig. 1(c)). A simpler and often effective approach is to use a null prompt ($\mathbf{c}_{\text{inv}} = \varnothing$) as the default. This inverts the mixed signal based on the model's general audio understanding without imposing specific content constraints during the inversion phase. The effect of $\mathbf{c}_{\text{inv}}$ and $\mathbf{c}_{\text{rev}}$ can be found in Tab. 9.

– **The Crucial Role of Guidance Weight $\omega$:** A key discovery is that achieving separation hinges on setting the classifier-free guidance weight $\omega$ appropriately, specifically $\omega \leq 1$. This is counterintuitive to typical generative usage where high $\omega$ values (e.g., $\omega = 3.5$ for AudioLDM2 [Liu et al., 2024]) amplify the conditional signal for a strong generation. In our context, when using $\mathbf{c}_{\text{inv}} = \varnothing$:

  – Setting $\omega = 0$ removes the conditional influence, effectively leading to a reconstruction of the original mixed audio.
  – Setting $\omega = 1$ removes the unconditional noise estimation from the combined prediction in Eq. (6), leaving only the component aligned with the target source described by $\mathbf{c}_{\text{rev}}$. This effectively isolates the target source during denoising.
  – Setting $\omega > 1$, as in standard generation, overly amplifies the conditional signal, leading to the synthesis of new content rather than the separation of existing audio components.

We empirically find that $\omega = 1$ yields the best separation results (as shown in Fig. 3(a)). This finding reveals that controlling the balance between conditional and unconditional predictions via $\omega$ is critical for steering the diffusion process from generation towards faithful separation.

By carefully selecting $\mathbf{c}_{\text{inv}}$, $\mathbf{c}_{\text{rev}}$, and setting $\omega$ (in practice, we set $\omega = 1$), we effectively repurpose the pre-trained audio diffusion model's generative capabilities to perform high-quality source separation without requiring any task-specific training.

### 3.4 Why Do T2A Diffusion Models Exhibit Zero-Shot Separation Ability?

The emergent source separation capability in Text-to-Audio (T2A) diffusion models is rooted in two core pillars: the **latent disentanglement** achieved during polyphonic training, and the **repurposing of the score function** via Classifier-Free Guidance (CFG) for an inverse problem.

**Latent Disentanglement via Data Compositionality.** T2A diffusion models are trained to estimate the probability density of multi-label, polyphonic audio (e.g., AudioSet [Gemmeke et al., 2017], with an average of $\sim 2.7$ concurrent labels per clip). To accurately generate a complex mixture, the model must implicitly learn the *compositional structure* of the acoustic environment. The model is forced to encode **disentangled latent factors** corresponding to individual sources $(s^{(1)}, \ldots, s^{(N)})$ such that their superposition correctly predicts the mixture $\mathbf{x}^{\text{mix}}$. This compositional training establishes an inherent, semantically-aligned structure in the latent space, where the representation of the mixture $\mathbf{x}^{\text{mix}}$ is a combination of the latent representations of its individual sources. This prerequisite knowledge makes the inverse task of decomposition feasible.

**The Score Function View and Guided Filtering.** The separation task is realized by repurposing the score-based generative process. The T2A model estimates the conditional score function via the noise prediction $\epsilon_\theta$, which is typically combined using Classifier-Free Guidance (illustrated in Eq. (6)). The procedure for separation involves two steps: first, the mixture audio $\mathbf{x}_{\text{mix}}$ is mapped into the noisy latent space via **DDIM Inversion**. Second, the key to separation is setting the guidance weight to $\omega = 1$. This condition simplifies the guidance to $\hat{\epsilon}_\theta(\mathbf{x}_t, \mathbf{c}) = \epsilon_\theta(\mathbf{x}_t, \mathbf{c})$, effectively removing the unconditional score component $\epsilon_\theta(\mathbf{x}_t, \varnothing)$ (which models the mixture's density) from the denoising process. By solely following the conditional score estimate, the model's reverse process is steered to follow the gradient direction leading to the data distribution consistent **only** with the target source defined by the text prompt $\mathbf{c}$. This mechanism thus transforms the generative model into a *zero-shot guided filtering* tool for source extraction.

## 4 Experiments

### 4.1 Experimental Settings

**Baselines.** To evaluate our training-free diffusion-based separation method, we compare it against two categories of existing approaches: **(i) Training-based methods.** These methods rely on large-scale supervised training and leverage text queries for targeted separation. We include: *LASS-Net* [Liu et al., 2022a], which conditions a mask estimator on text queries; *AudioSep* [Liu et al., 2023a], a scaled-up version of LASS-Net trained on massive multimodal data for zero-shot capabilities across diverse sources; and *FlowSep* [Yin et al., 2024], which enhances query-based separation using rectified continuous normalizing flows. We note that AUDIT [Wang et al., 2023] also uses audio diffusion models for instruction-guided audio editing (including source manipulation) but is not included as a direct baseline due to the lack of public code and data release for comparison. **(ii) Training-free methods.** These methods perform separation without requiring task-specific training data. We compare against: *NMF-MFCC* [Stöter et al., 2021], a classical non-negative matrix factorization approach operating on MFCC features for blind source separation; and *AudioEditor* [Manor and Michaeli, 2024], which achieves unsupervised separation by discovering principal components within the denoising process of a pre-trained diffusion model. In summary, training-based baselines require extensive annotated data for training, whereas other training-free baselines employ different underlying principles from our generative diffusion-based approach.

**Datasets.** We evaluate the open-set separation capabilities of our training-free method on two benchmark multimodal datasets with paired audio and text labels: The Audio–Visual Event (AVE) [Tian et al., 2018] dataset contains 4,143 video clips, each 10 seconds long, covering 28 distinct sound categories (e.g., *church bell*, *barking*, *frying*). AVE is valuable for evaluating separation in complex, real-world scenarios due to the presence of background noise, off-screen sounds, and varying event durations. The MUSIC dataset [Zhao et al., 2018] consists of clean solo performances from 11 musical instruments, thereby offering a controlled environment to assess the separation of individual, isolated sources with minimal interference. To facilitate comparison with prior research and ensure reproducibility, we use the official separation data splits for both AVE and MUSIC as provided by the DAVIS repository [Huang et al., 2024a].

Table 1: Main audio separation results comparing `ZeroSep` with training-based and training-free baselines on the AVE [Tian et al., 2018] and MUSIC [Zhao et al., 2018] datasets. Metrics are reported on the test sets. ↑ indicates higher is better, ↓ indicates lower is better. The best results are **bold**. More metrics can be found in Appendix A.

| Method | MUSIC | | | | AVE | | | |
|---|---|---|---|---|---|---|---|---|
| | FAD ↓ | LPAPS ↓ | C-A ↑ | C-T ↑ | FAD ↓ | LPAPS ↓ | C-A ↑ | C-T ↑ |
| *Require Separation Training* | | | | | | | | |
| LASS-Net | 1.039 | 5.602 | 0.204 | 0.014 | 0.626 | 6.062 | 0.232 | 0.011 |
| AudioSep | 0.725 | 5.209 | 0.450 | 0.204 | 0.446 | 5.733 | 0.457 | 0.167 |
| FlowSep | 0.402 | 5.578 | 0.564 | 0.245 | **0.258** | 4.719 | **0.493** | 0.082 |
| *Separation Training Free* | | | | | | | | |
| NMF-MFCC | 1.286 | 5.618 | 0.239 | −0.055 | 1.246 | 5.851 | 0.174 | **0.211** |
| AudioEdit | 0.568 | 4.869 | 0.453 | 0.196 | 0.372 | 4.959 | 0.341 | 0.074 |
| ZeroSep (Ours) | **0.377** | **4.669** | **0.615** | **0.271** | 0.269 | **4.537** | 0.442 | −0.001 |

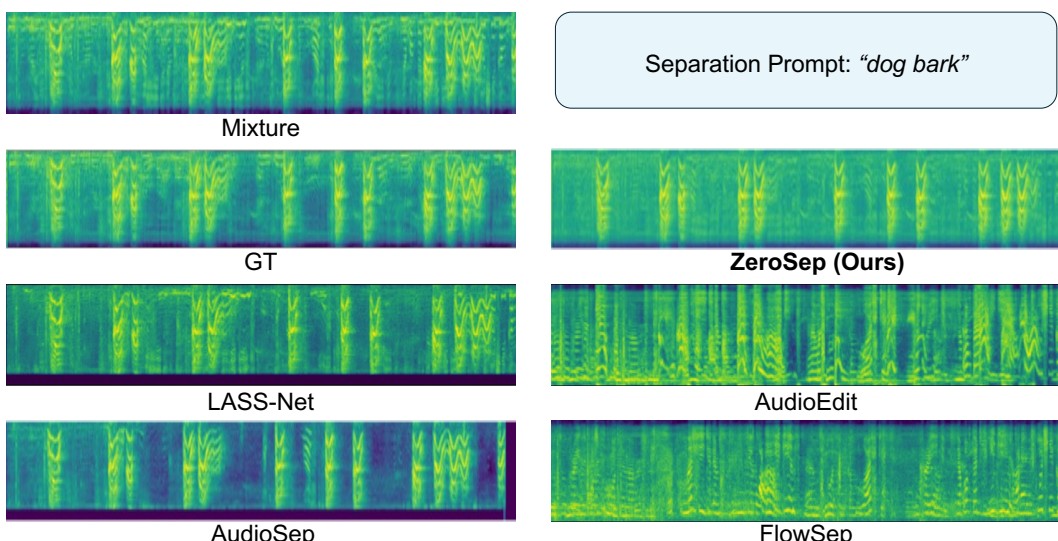

Figure 2: Qualitative visualization of audio separation results. The figure shows the input mixture (containing speech and dog barking) and the separated "dog barking" source produced by different baselines and `ZeroSep`. `ZeroSep`, guided by the text prompt "dog bark", successfully isolates the target sound, demonstrating its effectiveness compared to baseline methods. More separation results can be found in the appendix.

**Evaluation Metrics.** Traditional separation metrics–Signal-to-Distortion Ratio (SDR), Signal-to-Interference Ra tio (SIR), and Signal-to-Artifact Ratio (SAR) [Raffel et al., 2014]–quantify sample-level differences between a separated output $\hat{s}$ and the ground truth $s$. These metrics assume the output lies on the same waveform manifold as $s$, an assumption violated by generative models that may produce perceptu-

| Method | SDR | SIR | SAR |
|---|---|---|---|
| Original | 0.31 | 0.31 | 149.90 |
| VAE + Vcoder | −23.07 | −0.79 | −19.09 |

Table 2: Breakdown of SDR/SIR/SAR (with respect to the individual target source), for a generative reconstruction of the mixture versus the original mixture.

ally accurate but sample-wise divergent signals. Tab. 2 demonstrates how a VAE–Vocoder reconstruction of the mixture yields misleadingly poor SDR/SIR/SAR scores. Such an issue was previously studied in [Jayaram and Thickstun, 2020].

To capture perceptual and semantic fidelity of generative separation, we adopt metrics in embedding spaces: **Frechét Audio Distance (FAD)** [Kilgour et al., 2018]: measures the distance between embedding distributions of separated and ground-truth audio. **Learned Perceptual Audio Patch**

Table 3: Evaluation of AudioLDM [Liu et al., 2024], AudioLDM2 [Liu et al., 2024], and Tango [Ghosal et al., 2023] on the MUSIC and AVE benchmarks. We compare two U-Net sizes for each AudioLDM variant-S (181 M) / L (739 M) and AudioLDM2-S (350 M) / AudioLDM2-L (750 M), and Tango's 866 M-parameter U-Net. Results are reported for both DDIM and DDPM inversion methods, and for AudioLDM2 we include full vs. music-only training data.

| Model | Size | Data | MUSIC | | | | AVE | | | |
|---|---|---|---|---|---|---|---|---|---|---|
| | | | FAD↓ | LPAPS↓ | C-A↑ | C-T↑ | FAD↓ | LPAPS↓ | C-A↑ | C-T↑ |
| **DDIM Inversion** | | | | | | | | | | |
| AudioLDM | S | Full | 0.460 | 4.690 | 0.562 | 0.284 | 0.275 | 4.821 | 0.484 | 0.114 |
| | L | Full | 0.470 | 4.625 | 0.577 | 0.260 | 0.253 | 4.742 | 0.490 | 0.102 |
| AudioLDM2 | S | Full | 0.421 | 4.630 | 0.575 | 0.261 | 0.251 | 4.560 | 0.477 | 0.039 |
| | S | Music | 0.439 | 4.620 | 0.584 | 0.259 | 0.325 | 4.666 | 0.424 | 0.106 |
| | L | Full | 0.377 | 4.669 | 0.615 | 0.271 | 0.269 | 4.537 | 0.442 | −0.001 |
| Tango | L | Full | 0.606 | 4.511 | 0.544 | 0.204 | 0.724 | 4.451 | 0.437 | 0.077 |
| **DDPM Inversion** | | | | | | | | | | |
| AudioLDM | S | Full | 0.417 | 4.580 | 0.605 | 0.300 | 0.239 | 4.681 | 0.504 | 0.133 |
| | L | Full | 0.388 | 4.536 | 0.626 | 0.283 | 0.266 | 4.629 | 0.496 | 0.108 |
| AudioLDM2 | S | Full | 0.390 | 4.586 | 0.595 | 0.238 | 0.272 | 4.546 | 0.488 | 0.041 |
| | S | Music | 0.384 | 4.596 | 0.609 | 0.259 | 0.238 | 4.628 | 0.467 | 0.126 |
| | L | Full | 0.397 | 4.581 | 0.598 | 0.239 | 0.267 | 4.523 | 0.445 | −0.008 |
| Tango | L | Full | 0.539 | 4.474 | 0.581 | 0.189 | 0.723 | 4.471 | 0.451 | 0.032 |

**Similarity (LPAPS)** [Manor and Michaeli, 2024]: evaluates perceptual audio similarity in a learned embedding space. **CLAP-A** and **CLAP-T** [Yin et al., 2024]: CLAP-A is the cosine similarity between audio embeddings of the separation output and the ground-truth source; CLAP-T is the cosine similarity between audio embeddings and the text embedding of the target class. These feature-based metrics better reflect perceptual quality and semantic alignment, addressing the shortcomings of waveform-level reference metrics for generative audio separation. We further assess whether the model truly separates the original source content or merely generates audio with similar semantic qualities that may not remain consistent with the ground-truth mixture (Appendix A).

## 4.2 Main Comparison

Tab. 1 presents the core results of our evaluation, comparing the performance of our training-free method, ZeroSep, against representative training-based and other training-free baselines on the AVE and MUSIC datasets. Remarkably, ZeroSep demonstrates performance that surpasses the leading supervised methods, effectively challenging the necessity of large-scale supervised training for state-of-the-art audio separation. On the MUSIC dataset, ZeroSep outperforms the strongest training-based baseline FlowSep across all metrics. On the more complex and open-domain AVE dataset, ZeroSep achieves performance comparable to FlowSep.

The training-based baselines show a clear improvement trend with increasing model size and data: LASS-Net is surpassed by AudioSep, which in turn is surpassed by FlowSep, underscoring the benefits of extensive supervised training data and better models. The other training-free methods evaluated, NMF-MFCC and AudioEditor, yield substantially lower performance than the top supervised methods, highlighting the difficulty of achieving high-quality separation without leveraging separation training, which ZeroSep has addressed.

Beyond quantitative scores, the qualitative visualization in Fig. 2 provides further evidence of ZeroSep's effectiveness, illustrating the successful separation of a target sound (e.g., "dog barking") from a complex mixture containing other sources like human speech. These results collectively indicate that pre-trained text-guided diffusion models possess powerful inherent capabilities that can be effectively harnessed for audio separation without the need for task-specific training.

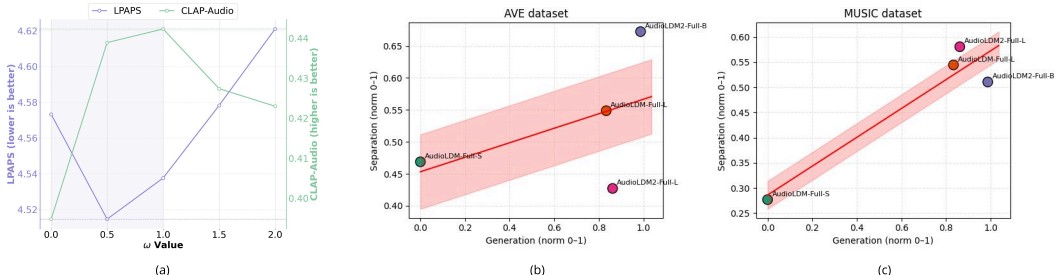

Figure 3: (a) Impact of guidance weight $\omega$: increasing $\omega$ from 0 to 1 improves separation metrics (LPAPS and CLAP-A), whereas $\omega > 1$ degrades performance below the mixture baseline ($\omega = 0$), underscoring the critical role of $\omega$. (b)–(c) Positive correlation between separation quality (normalized all scores from Tab. 3) and generative capability (normalized FAD scores on AudioCap [Liu et al., 2023b], [Liu et al., 2024]) across AudioLDM variants, indicating that stronger generation can potentially lead to better separation.

## 4.3 Ablation Studies

In this section, we analyze the influence of various components on `ZeroSep`'s separation performance to identify factors contributing to its effectiveness. We investigate aspects including the choice and capacity of the base generative model, the impact of its training data domain, inversion strategies, guidance weight effects, and prompt selection. More analysis can be found in the appendix.

**How Does the Base Generative Model Affect Separation?** Since `ZeroSep` is built upon a pre-trained diffusion model, understanding how this base model affects separation is crucial. First, we compare separation performance using different base model architectures, including models from the AudioLDM [Liu et al., 2023b], AudioLDM2 [Liu et al., 2024], and Tango [Ghosal et al., 2023] families, as shown in Tab. 3. The results indicate that various base models can yield separation performance comparable to the best training-based baseline, FlowSep, demonstrating versatility in base model selection. Specifically, models from the AudioLDM and AudioLDM2 families generally outperform Tango in this separation task.

Second, we analyze the effect of model capacity by comparing different sizes within the AudioLDM and AudioLDM2 families (e.g., AudioLDM-S vs. AudioLDM-L, AudioLDM2-S vs. AudioLDM2-L). As shown in Tab. 3, increasing model size consistently leads to improved separation performance. This suggests a positive correlation between the generative power of the base model and its effectiveness for separation. We further visualize this trend by plotting the correlation between generative performance (measured by FAD) and separation metrics in Fig. 3(b) and (c), which confirms that stronger generative models tend to yield better separation results.

Third, we investigate the impact of the base model's training data domain. Tab. 3 includes results for a model trained specifically on MUSIC data compared to the same model trained on a broader data corpora. We observe that the MUSIC-data-trained model achieves performance on the MUSIC dataset that is similar to or even better than the full-training model for certain metrics (e.g., LPAPS and CLAP-A). This finding suggests that if the target separation domain is narrow, a base generative model trained on domain-specific data can be sufficient or even advantageous, potentially increasing the separation flexibility.

**Inversion Strategy.** As shown in Tab. 3, both DDIM and DDPM inversion methods enable competitive separation performance relative to the baselines. Analyzing their behavior across different base model capacities and training data domains, we observe that DDPM inversion tends to yield more stable metrics, exhibiting less fluctuation with respect to changes in model size and training data. In contrast, DDIM inversion shows larger variations under these different conditions. This analysis indicates that `ZeroSep`'s effectiveness is not strictly tied to a single inversion technique, offering flexibility in implementation.

**Effect of** $\omega$**.** As detailed in Sec. 3.3, setting $\omega = 0$ effectively reduces the process to an unconditional reconstruction, while higher values increase the adherence to the text prompt $\mathbf{c}_{\text{rev}}$. We analyze the impact of $\omega$ on separation performance by evaluating values in the set $\{0, 0.5, 1, 1.5, 2\}$. Fig. 3(a) presents the results for LAPAS and CLAP-A metrics. It can be observed that as $\omega$ increases from

Table 4: Effect of $\mathbf{c}_{\mathrm{inv}}$ and $\mathbf{c}_{\mathrm{rev}}$ on separation metrics. Triangles indicate change relative to the baseline, with ▼ denoting improvement and ▼ denoting degradation.

| $\mathbf{c}_{\mathrm{inv}}$ | $\mathbf{c}_{\mathrm{rev}}$ | MUSIC | | | | AVE | | | |
|---|---|---|---|---|---|---|---|---|---|
| | | FAD ↓ | LPAPS ↓ | C-A ↑ | C-T ↑ | FAD ↓ | LPAPS ↓ | C-A ↑ | C-T ↑ |
| $\varnothing$ | $\mathbf{c}^{(i)}$ | 0.377 | 4.669 | 0.615 | 0.271 | 0.269 | 4.537 | 0.442 | −0.001 |
| $\varnothing$ | random | 0.577 | 4.900 | 0.363 | 0.125 | 0.325 | 4.749 | 0.289 | 0.019 |
| | | ▲ 0.200 | ▲ 0.231 | ▼ 0.252 | ▼ 0.146 | ▲ 0.056 | ▲ 0.212 | ▼ 0.153 | ▲ 0.020 |
| $\mathbf{c}^{(j)}$ | $\mathbf{c}^{(i)}$ | 0.454 | 4.547 | 0.581 | 0.254 | 0.321 | 4.599 | 0.496 | 0.055 |
| | | ▲ 0.077 | ▼ 0.122 | ▼ 0.034 | ▼ 0.017 | ▲ 0.052 | ▲ 0.062 | ▲ 0.054 | ▲ 0.056 |

0 to 1, both separation metrics generally improve, indicating that conditioning on the target source prompt effectively guides the separation. However, beyond $\omega = 1$, performance deteriorates sharply, suggesting that excessively strong guidance can lead to suboptimal reconstructions or introduce artifacts. Based on this analysis, we empirically set $\omega = 1$ for our main experiments to achieve the best balance between adherence to the target prompt and reconstruction quality.

**Effect of $\mathbf{c}_{\mathrm{rev}}$ and $\mathbf{c}_{\mathrm{inv}}$.** Tab. 9 summarizes the separation performance under different prompt configurations. First, replacing the prompt $\mathbf{c}_{\mathrm{rev}}$ specifying the target sound source with a random, mixture-unrelated prompt results in a drastic performance drop across all metrics. This highlights the essential role of accurate text conditioning towards separating target source. For $\mathbf{c}_{\mathrm{inv}}$, we explore replacing the null prompt with a prompt for a different source present in the mixture but not the target. This substitution leads to a slight degradation in performance, which demonstrates that while $\mathbf{c}_{\mathrm{inv}}$ provides some contextual information, the method is less sensitive to its precise content.

## 5   Conclusion

This paper demonstrates a new paradigm for audio source separation, moving away from reliance on extensive supervised training. In particular, we introduce `ZeroSep`, a novel training-free approach that leverages the power of pre-trained text-guided audio diffusion models. Our evaluation reveals that `ZeroSep` achieves performance on par with or exceeds leading supervised separation baselines across benchmark datasets, and our analysis further illuminates the factors critical for successful transformation from generation to separation. The effectiveness of `ZeroSep` showcases a new application for the growing family of audio diffusion models and offers a compelling alternative direction for developing open-set audio source separation models.

**Limitations.** While we have demonstrated the efficacy of `ZeroSep` on popular audio diffusion models (e.g., AudioLDM families and Tango), how it works on larger models and alternative architectures remains untested. In addition, our reliance on latent inversion can introduce approximation errors that may impair separation fidelity. Due to computational constraints, we did not include these experiments in this work. We will explore how to scale evaluations to diverse, high-capacity diffusion models and develop more accurate inversion techniques in future work.

**Beyond Basic Separation.** `ZeroSep`'s inherent mechanism unlocks diverse application scenarios beyond simple text-to-sound separation. First, text prompts can be automatically generated. Audio event detection [Mesaros et al., 2021] or audio language models [Gong et al., 2023, Ghosh et al., 2024] can derive labels or free-form descriptions from mixtures, enabling automated separation. Second, `ZeroSep` facilitates cross-modal applications; for instance, leveraging audio-visual localization [Huang et al., 2023, Chen et al., 2021b] and vision-language models [Liu et al., 2023c], users could separate sounds in a video by visually describing sounding objects. Third, recognizing the importance of spatial audio understanding and rendering [Gao and Grauman, 2019, Liang et al., 2023a,b, Huang et al., 2024b] for human-level acoustic perception, `ZeroSep` can be directly extended to spatial audio separation using diffusion models that support multi-channel input, such as Stable Audio Open [Evans et al., 2025b]. Finally, our method naturally enables a continuous transition from mixture reconstruction to sound highlighting [Gandikota et al., 2024a, Huang et al., 2024c, 2025a,b] by varying the value of classifier-free guidance, allowing scaling of target sound elements from full presence to complete separation [Gandikota et al., 2024b].

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

Table 5: Additional evaluation on the AVE [Tian et al., 2018] and MUSIC [Zhao et al., 2018] datasets with MSE and Encodec-based metrics. ↓ indicates lower is better. The best results are **bold**.

| Method | MUSIC | | | AVE | | |
|---|---|---|---|---|---|---|
| | MSE ↓ | L1 (Emb.) ↓ | L2 (Emb.) ↓ | MSE ↓ | L1 (Emb.) ↓ | L2 (Emb.) ↓ |
| *Require Separation Training* | | | | | | |
| LASS-Net | 32.860 | 0.6205 | 0.7338 | 12.612 | 0.5984 | 0.6737 |
| AudioSep | 10.857 | **0.4232** | **0.4046** | 9.383 | **0.4949** | **0.5095** |
| FlowSep | 15.134 | 0.5525 | 0.5869 | **7.785** | 0.6706 | 0.8402 |
| *Separation Training Free* | | | | | | |
| NMF-MFCC | 36.761 | 0.4230 | 0.4180 | 59.594 | 0.5766 | 0.6906 |
| AudioEdit | 29.357 | 0.8059 | 1.3088 | 22.253 | 0.9588 | 1.7090 |
| ZeroSep (Ours) | **15.470** | 0.5540 | 0.5912 | 11.076 | 0.7959 | 1.1258 |

# A  More Metrics

To further assess separation quality, we report additional metrics, including L1/L2 loss on Encodec [Défossez et al., 2022] embeddings and MSE on mel-spectrograms, which capture temporal fidelity more effectively. Results are shown in Tab. 5. On these newly introduced metrics, ZeroSep achieves performance comparable to the supervised FlowSep model and surpasses LASS-Net on the MUSIC dataset, providing stronger evidence of genuine source separation rather than prompt-consistent hallucination.

An interesting observation arises with NMF-MFCC. Unlike other methods, which typically fail by leaving residual interference, NMF-MFCC often produces silence or overly smoothed outputs. This distinct failure mode can artificially benefit the Encodec embedding distances: the smoothed, low-detail outputs align more closely with Encodec's coarse embedding space, despite discarding much of the fine-grained content. As a result, NMF-MFCC appears competitive on L1/L2 embedding metrics, yet its poor performance on perceptual and semantic measures (e.g., FAD, LPAPS, and mel-spectrogram MSE) highlights its limitations.

# B  Effect of $\omega$ and Scheduled Guidance

Large guidance weights $\omega$ amplify the conditional score. Since both the unconditional and conditional scores are approximations, excessive scaling can push samples into low-probability regions off the data manifold, improving prompt adherence and sample diversity but risking artifacts—a phenomenon broadly reported in image/audio diffusion models [Ho and Salimans, 2022, Sadat et al., 2024]. Consequently, large $\omega$ favors generative hallucination rather than accurate separation. Empirically, we find $\omega = 1$ strikes a good balance between suppressing interfering sounds and avoiding spurious generations.

**Static vs. Scheduled Guidance.** Intuitively, early diffusion steps shape the global structure, while later steps refine details. We therefore compared constant and scheduled $\omega$ on the AudioLDM2-Large backbone (DDIM, 50 steps). Results on MUSIC are shown in Tab. 6.

Table 6: Comparison of static vs. scheduled guidance ($\omega$) on MUSIC.

| $\omega$ schedule | FAD ↓ | LPAPS ↓ | C-A ↑ | C-T ↑ |
|---|---|---|---|---|
| constant 1 | 0.377 | 4.669 | 0.615 | 0.271 |
| linear 0→1 | 0.471 | **4.526** | 0.520 | 0.202 |
| linear 1→0 | **0.332** | 4.667 | **0.618** | **0.281** |
| sine 0→1 | 0.431 | 4.523 | 0.550 | 0.227 |

**Observations.** (i) Low-to-high schedules (0→1, sine) degrade separation, as early under-conditioning causes loss of clean target structure. (ii) High-to-low schedules (1→0) improve over constant $\omega = 1$, consistent with reports that guidance is most useful in the early-to-mid noise range

but less helpful at the end. This supports our main claim that $\omega = 1$ is an effective default while also motivating dynamic scheduling as a promising future direction.

## C  Runtime and Scalability for Multiple Sources

Let $S$ denote the number of inference steps and $T_d$ the per-step time. `ZeroSep` requires a single inversion of the mixture (shared for all branches) plus one guided denoising per target source:

$$T_{\text{ZeroSep}} \approx \underbrace{S\,T_d}_{\text{inversion}} + \underbrace{N\,S\,T_d}_{N\text{ targets}} = (N+1)S\,T_d.$$

Discriminative models (LASS-Net, AudioSep) require only a single forward pass per target ($N\,T_d$). FlowSep, a rectified-flow model, requires multi-step denoising ($N\,S\,T_d$). Tab. 7 summarizes complexity and measured runtimes.

Table 7: Runtime complexity and average cost (separating one source) on A100 GPU.

| Method | Complexity | Runtime (s) |
|---|---|---|
| LASS-Net | $N\,T_d$ | 0.05 |
| AudioSep | $N\,T_d$ | 0.04 |
| FlowSep | $N\,S\,T_d$ | 1.07 |
| `ZeroSep` (AudioLDM-S) | $(N+1)S\,T_d$ | 0.98 |

While `ZeroSep` is slower than discriminative models, it matches FlowSep. Crucially, runtime scales down with advanced samplers: DPM-Solver++ attains high quality in 15–20 steps, and consistency models in 2–8 or even 1 step, which would directly reduce `ZeroSep` 's runtime.

## D  Polyphonic Mixtures

`ZeroSep` naturally extends to multi-source separation. On 3-source MUSIC mixtures (e.g., violin+flute+trumpet), results degrade moderately compared to 2-source (8).

Table 8: `ZeroSep`'s performance on 2- and 3-source mixtures from the MUSIC dataset.

| # Sources | FAD ↓ | LPAPS ↓ | C-A ↑ | C-T ↑ |
|---|---|---|---|---|
| 2 | 0.377 | 4.669 | 0.615 | 0.271 |
| 3 | 0.508 | 4.248 | 0.501 | 0.246 |

For especially challenging cases where sources share the same event class, attribute-qualified prompts (e.g., *"soft violin arpeggio"* vs. *"long sustained violin note"*) may help.

## E  Prompt Sensitivity and Hierarchical/Open-Set Behavior

We tested prompt robustness by perturbing MUSIC class names with hypernyms (e.g., *violin* → *string instrument*) and underspecified descriptors (e.g., *trumpet* → *bright instrument*). Results are summarized in Tab. 9.

Table 9: Prompt sensitivity analysis on MUSIC.

| Prompt type | FAD ↓ | LPAPS ↓ | C-A ↑ | C-T ↑ |
|---|---|---|---|---|
| canonical | 0.377 | 4.669 | 0.615 | 0.271 |
| hypernym | 0.388 | 4.692 | 0.583 | **0.280** |
| underspecified | **0.336** | 4.714 | 0.577 | 0.248 |

**Observations.** Hypernyms preserve family-level correctness, slightly degrading accuracy but improving consistency (C-T). Underspecified descriptors sometimes improve distributional metrics (FAD) while reducing class accuracy, indicating plausible but less precise extractions. These findings suggest hierarchical robustness: `ZeroSep` generalizes reasonably from coarse categories (e.g., "string instrument") to specific instances (e.g., violin), though we avoid over-claiming open-set separation.

Table 10: Effect of prompt descriptiveness on MUSIC.

| Prompt type | FAD ↓ | LPAPS ↓ | C-A ↑ | C-T ↑ |
|---|---|---|---|---|
| Class label | **0.377** | 4.669 | **0.615** | **0.271** |
| Descriptive | 0.484 | **4.541** | 0.512 | 0.196 |
| AudioSep | 0.725 | 5.209 | 0.450 | 0.204 |

## F  Prompt Length and Descriptiveness

We conducted a focused study on the MUSIC dataset to test robustness to descriptive prompts (shown in Tab. 10). Each of the 11 instrument class labels was expanded into a free-form caption roughly 10× longer than the label. For example, *"saxophone"* was expanded to *"warm, breathy reed tone with slight rasp and expressive bends"*, and *"accordion"* to *"reedy, wheezy sustained chords with gentle tremolo and slow swells"*. This setting stresses `ZeroSep` with more detailed but potentially noisy prompts.

**Observations.** `ZeroSep` shows a moderate degradation under long, descriptive prompts but still outperforms the strong supervised baseline AudioSep across most metrics. We attribute this drop to a mismatch between our prompt expansions and the captions used to train text-to-audio diffusion models, which are typically short, coarse descriptions (e.g., *"a man speaking while a baby is crying"*). This highlights a promising direction: aligning prompt style with training data or finetuning on more descriptive captions to better exploit natural-language expressiveness.

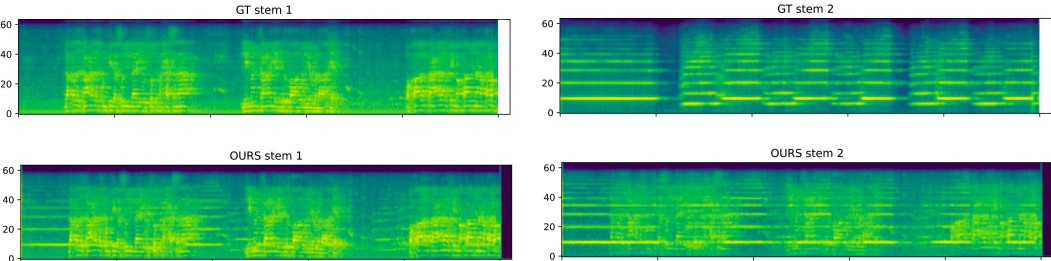

Figure 4: Failure case analysis of `ZeroSep`. Mixture: Man speech (stem 1) + Shofar (stem 2).

## G  Failure Case Analysis

While generally effective, `ZeroSep` can sometimes fail to fully isolate the target source. This typically occurs when an interfering source possesses significant energy that the model cannot eliminate in a single iteration. An illustrative example of such a failure is presented in Fig. 4. We postulate that, given the inherent progressive operation of diffusion models, the removal of interfering sources also proceeds incrementally. Consequently, this performance limitation may be tied to the number of inference steps utilized. Potential avenues for improvement include increasing the inference steps or iteratively applying the separation process.

## H  More Separation Results

These figures present mel-spectrograms that visualize the audio separation performance on two-source mixtures. For each figure, the rows are ordered from top to bottom as follows: the first source's Ground Truth, followed by its separation results from LASS-Net, FlowSep, AudioEdit, AudioSep,

and Ours. This sequence is then repeated for the second source: Ground Truth 2, LASS-Net 2, FlowSep 2, AudioEdit 2, AudioSep 2, and Ours 2. You might notice some white or empty areas on the right side of the mel-spectrograms; these are simply due to the varying lengths of the audio samples.

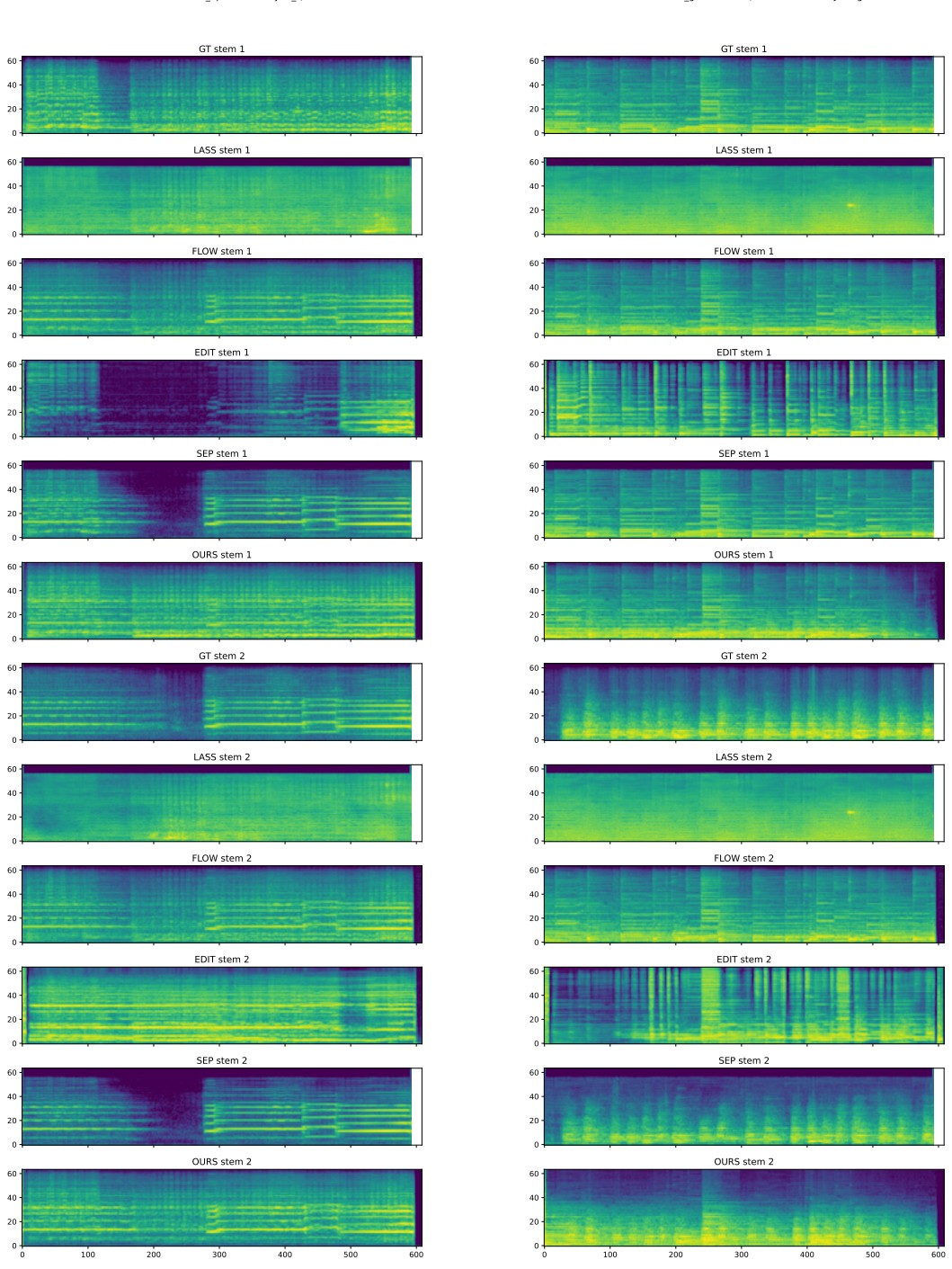

Figure 5: Mixture: Cello (stem 1) + Erhu (Stem 2)

Figure 6: Mixture: Acoustic Guitar (stem 1) + Tuba (Stem 2)

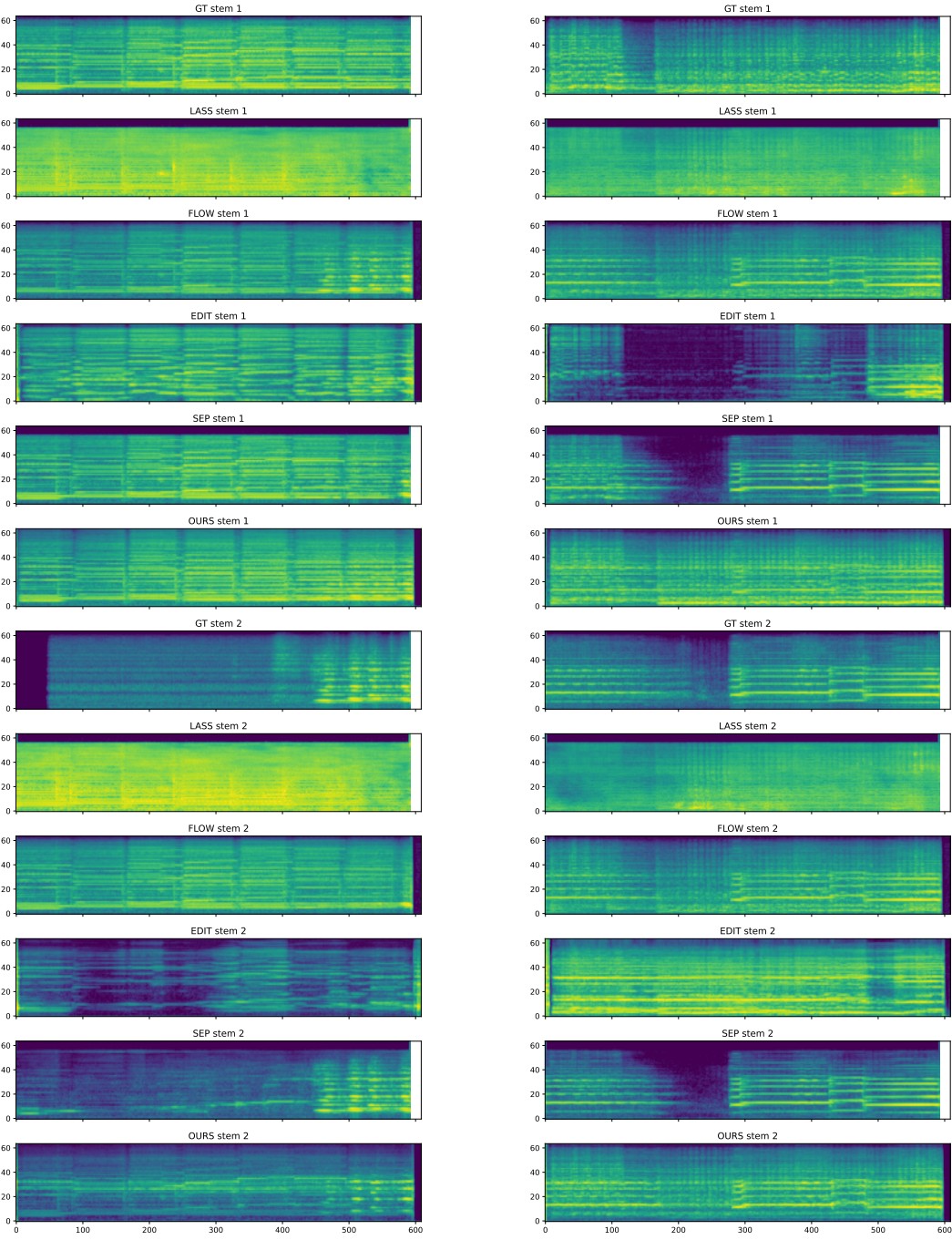

Figure 7: Mixture: Accordion (stem 1) + Flute (Stem 2)

Figure 8: Mixture: Cello (stem 1) + Erhu (Stem 2)

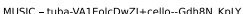

MUSIC – tuba-VA1EolcDwZI+cello--Gdh8N_KpLY

MUSIC – xylophone-5lm9laLSORc+trumpet-I2QXo4mGeRE

GT stem 1

LASS stem 1

FLOW stem 1

EDIT stem 1

SEP stem 1

OURS stem 1

GT stem 2

LASS stem 2

FLOW stem 2

EDIT stem 2

SEP stem 2

OURS stem 2

Figure 9: Mixture: Tuba (stem 1) + Cello (Stem 2)

Figure 10: Mixture: Xlyophone (stem 1) + Trumpet (Stem 2)

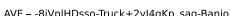

AVE − -8jVnlHDsso-Truck+2vJ4gKp_sag-Banjo

AVE − -9ummbDsgFM-Chainsaw+1NYCiPBzn-E-Accordion

GT stem 1

LASS stem 1

FLOW stem 1

EDIT stem 1

SEP stem 1

OURS stem 1

GT stem 2

LASS stem 2

FLOW stem 2

EDIT stem 2

SEP stem 2

OURS stem 2

Figure 11: Mixture: Truck (stem 1) + Banjo (Stem 2)

Figure 12: Mixture: Chainsaw (stem 1) + Accordion (Stem 2)

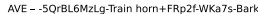

GT stem 1

LASS stem 1

FLOW stem 1

EDIT stem 1

SEP stem 1

OURS stem 1

GT stem 2

LASS stem 2

FLOW stem 2

EDIT stem 2

SEP stem 2

OURS stem 2

Figure 13: Mixture: Train Horn (stem 1) + Bark (Stem 2)

GT stem 1

LASS stem 1

FLOW stem 1

EDIT stem 1

SEP stem 1

OURS stem 1

GT stem 2

LASS stem 2

FLOW stem 2

EDIT stem 2

SEP stem 2

OURS stem 2

Figure 14: Mixture: Male Speech (stem 1) + Airplane (Stem 2)

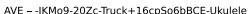

Figure 15: Mixture: Truck (stem 1) + Ukulele (Stem 2)

Figure 16: Mixture: Bark (stem 1) + Toilet Flush (Stem 2)

