# OpenReview forum: "ZeroSep: Separate Anything in Audio with Zero Training"
_NeurIPS.cc/2025/Conference — NeurIPS 2025 poster_

### Official Review · Reviewer_V2hC · 2025-06-22

**Clarity:** 3
**Significance:** 3
**Originality:** 2
**Rating:** 4
**Confidence:** 5

**Summary:**

In this paper the authors propose a training-free way of doing audio source separation via a diffusion model, without requiring the user to re-train or fine-tune the diffusion model for the particular separation task (like LASS-Net / AudioSep).

The method first inverts the observed variable (the mixture $x^{mix}$) to the latent domain via the forward process (additionally given a conditional text embedding $c_{inv}$ describing the mixture). Then, given the resulting noisy latent $x_T$ one does the reverse process with classifier free guidance, setting the $c_{rev}$ prompt to describe a source in the mix.

The authors show that the best results are obtained with classifier free guidance weighting of 1. The authors also ablate the choice of the diffusion model, the inversion strategy (DDIM inversion vs DDPM inversion), the effect of the classifier free guidance weight, and the type of text embedding $c_{inv}, c_{rev}$ to use during inversion and conditional inference, respectively.

**Questions:**

- Line 125: Bold subscripts $\textbf{0}$ an $\textbf{T}$ while following text uses normal font.
- Line 127: "parameterized by θ" should be "parameterized by a set of weights \theta".
- Line 128: "reversing the diffusion" should be "reversing the diffusion process".
- Line 128: If one assume using AudioLDM2, one first has to decode the final latents to obtain the mel-spectrograms. Also, there are many models (e.g., Stable Audio Open) which do not operate over mel-spectrograms.
- Line 155: Here again the latent space doesn't have nothing to do with mel-spectraograms. Specify this is your particular setting.
- Line 163: Audio source separation is not an inherently discriminative task. It can be called discriminative if one uses discriminative models to solve the task. The presented method and many other works effectively engage in generative source separation. See this recent survey for a detailed listing of works in generative source separation: https://arxiv.org/abs/2506.08457v1 (references [247]–[253]).

**Ethical Concerns:**

["NO or VERY MINOR ethics concerns only"]

**Final Justification:**

I decided to raise the score given that authors have addressed my concerns (such modifications should appear in the camera ready version), namely adjusting the novelty claim, adding missing references, stating in what way the method differs from AudioEditor and integrating L1 / L2 embedding metrics in the quantitative evaluation.

**Limitations:**

yes

**Quality:**

3

**Strengths And Weaknesses:**

**Strengths**

- The authors apply successfully DDIM and DDPM inversion for solving source separation via diffusion without requiring to have a pre-trained audio model for this specific task. Solving this task zero-shot via a powerful pre-trained text-conditioned diffusion model feels a natural direction, instead of having to retrain (e.g., LASS-Net / AudioSep), not taking advantage of an available prior model.

**Weaknesses**

**Missing citations**

- First and foremost, there are many important citations that the paper is missing. For example:
  - It does not cite modern latent diffusion models such as the state-of-the-art Stable Audio Open (Evans et al. 2025) which greatly improves over AudioLDM2.
  - It does not cite prominent previous baselines for performing source separation with diffusion models like SepDiff (Chen et al. 2023) and DiffSep (Scheibler et al. 2023). Those were the two first papers solving source separation via diffusion models, so citing them in a paper about source separation via diffusion models should be the minimum. To be even more correct, the first model to use score-based methods for source separation was (Jayaram et al. 2020), appearing even before the release of the DDPM paper dated 19 June 2020 (see https://arxiv.org/abs/2006.11239v1), and was based on the first formulation of Yang Song of denoising score-matching employing noise-conditioned score networks (NCSNs) (Song et al. 2019); more about the paper of Jayaram et al. in the following).
  - The authors do not cite the highly relevant DITTO baseline (Novack et al. 2024) that also does inference-only editing without modifying the base diffusion model.

**Novelty overclaim**

- It is not true that ZeroSep is the *"first training-free audio source separation framework repurposing pre-trained text-guided audio diffusion models"* as the paper claims (verbatim). Generalized Multi-Source Diffusion Inference (GMSDI) (Postolache et al. 2024) sets out to acomplish the same task (to be precise more tasks, also including accompaniment generation) with a pre-trained text-guided audio diffusion model without any custom training or fine-tuning for the task. That method differs from the present paper since it uses posterior sampling with a Dirac-based likelihood function instead of a inversion mechanism, but it does accomplish literally the same task with the same assumptions on not pre-training / fine-tuning the base diffusion model for the separation task.
- Also regarding the technical contribution:
   1. The methodology is very similar to the text version of *AudioEditor*, which the authors compare to, albeit in the unsupervised modality. The authors of AudioEditor presented a text-based inversion method for audio that largely overlaps with the proposed methodology, albeit being applied to editing instead of source separation, where it is applied in this paper.
   2. Relating the GMSDI paper, a very similar principle is presented for source extraction in (Eq. 14) (*GMSDI extractor*). While doing posterior sampling instead of inversion, the user only has to provide a description of the mixture and a description of the source to be extracted. While the authors compare with training free methods, those are not text-guided like GMSDI. The authors should compare with such text-based posterior sampling, keeping the underlying generative model fixed, in order to show that their methodology improves over it.
  3. Understanding that CFG weighting is crucial is not something surprising, since it is well known that setting $w= 0$ removes conditioning and by setting it too high results degrade (for example see Fig. 8 in (Karras et al. 2024) and Table 1 in (Postolache et al. 2024)).

**Concerning metrics**

First thing, the inadequacy of standard SS metrics like SDR illustrated in Table 3, was previously studied in (Jayaram et al. 2020), see Figure 4 (I believe this merits a citation). They also proposed to use FID as a distribution-based metric (they were doing image source separation in that paper). Regarding FAD, the authors do not detail which embedding space is used for computing FAD (we have vggish, Encodec, CLAP, etc. and they work in different ways (Gui et al. 2024)).

I do not think that the FAD metric (which measure semantic similarity on sets), together with the CLAP / LPAPS metrics
(which encode all temporal content in a global compressed vector) are sufficient to quantify if we are really separating the content or we are only generating something with the same semantic qualities which, however could not be consistent with the ground truth source in the mixture. For example, doing an L1 / L2 loss on Encodec embeddings could give something more robust An even better metric is the  MSE on mel-spectograms like in (Karchkhadze et al. 2025). They use the AudioLDM baseline (like the authors) for doing supervised source separation. They show improvement over a baseline in source separation, showcasing that such a metric is useful for measuring  source separation even when generating via a latent diffusion model.

Thus, Table 1, should be improved,  because the authors only match the semantic content at the moment, which is clearly enforced by the textual prompt during generation.

**Summary**

While the direction is positive, there are problems concerning (i) missing important citations (ii) novelty overclaim and (most importantly) (iii) the authors do not compute any metric that matches the outcomes with the ground-truths at a temporal level (they do only semantic level). As such, I am inclined to reject the paper at the moment; authors can improve their paper solving these three issues.

**References**

- Evans, Zach et al. “Stable Audio Open.” ICASSP 2025-2025 IEEE International Conference on Acoustics, Speech and Signal Processing (ICASSP). IEEE, 2025.
- B. Chen, C. Wu, and W. Zhao, “SEPDIFF: Speech separation based on denoising diffusion model,” in ICASSP, Jun. 2023, pp. 1–5.
- R. Scheibler, Y. Ji, S.-W. Chung, J. Byun, S. Choe, and M.-S. Choi, “Diffusion-based generative speech source separation,” in ICASSP, Jun. 2023, pp. 1–5.
- Jayaram, Vivek, and John Thickstun. "Source separation with deep generative priors." International Conference on Machine Learning. PMLR, 2020.
- Song, Yang, and Stefano Ermon. "Generative modeling by estimating gradients of the data distribution." Advances in neural information processing systems 32 (2019).
- Novack, Zachary, et al. "DITTO: Diffusion Inference-Time T-Optimization for Music Generation." Forty-first International Conference on Machine Learning. 2024.
- Hila Manor and Tomer Michaeli. Zero-shot unsupervised and text-based audio editing using DDPM inversion. In Ruslan Salakhutdinov, Zico Kolter, Katherine Heller, Adrian Weller, Nuria Oliver, Jonathan Scarlett, and Felix Berkenkamp, editors, Proceedings of the 41st International Conference on Machine Learning, volume 235 of Proceedings of Machine Learning Research, pages 34603–34629. PMLR, 21–27 Jul 2024.
- Postolache, Emilian, et al. "Generalized multi-source inference for text conditioned music diffusion models." ICASSP 2024-2024 IEEE International Conference on Acoustics, Speech and Signal Processing (ICASSP). IEEE, 2024.
- Karras, Tero, et al. "Guiding a diffusion model with a bad version of itself." Advances in Neural Information Processing Systems 37 (2024): 52996-53021.
- Gui, Azalea, et al. "Adapting frechet audio distance for generative music evaluation." ICASSP 2024-2024 IEEE International Conference on Acoustics, Speech and Signal Processing (ICASSP). IEEE, 2024.

---

> ### Author Rebuttal · Authors · 2025-07-30
>
> We sincerely thank you for the thorough, constructive review. Below we (i) add the missing citations, (ii) clarify novelty and positioning, (iii) extend the evaluation with time‑aligned metrics, and (iv) correct wording/formatting issues.; we will incorporate all changes in the revised manuscript.
>
> ---
>
> ### Q1 Missing Citations
> We thank the reviewer for providing the paper list to help us make a more comprehensive bibliography. We have added all requested references as followed:
>
> | Ref.                                                | Where added                                           |
> | --------------------------------------------------- | ----------------------------------------------------- |
> | Stable Audio Open (Evans ’25)                       | p3 L91‑92 (modern T2A backbones)                      |
> | SepDiff / DiffSep (Chen & Scheibler ’23)            | p3 L103‑106 (diffusion‑based separation)              |
> | Jayaram et al. ’20                                  | p3 L106 (earliest score‑based separation)             |
> | DITTO (Novack ’24)                                  | p3 L110‑112 (inference‑only editing)                  |
> | GMSDI (Postolache ’24)                              | p3 L108‑110  (zero‑shot text‑guided separation) |
>
>
> These changes ensure the paper accurately cites prior works.
>
> ---
>
> ### Q2. Novelty Clarification & Positioning
>
> We agree that our original phrasing overstated novelty.  We now state:
>
> > “ZeroSep is *a* training‑free audio source‑separation framework that repurposes pre‑trained text‑guided diffusion models…”
>
> **Difference  to GMSDI.**
> GMSDI (Postolache et al. 2024) performs zero‑shot separation, but via **posterior sampling with a Dirac likelihood**. ZeroSep instead performs **latent‑space inversion plus text‑conditioned denoising**. While GMSDI can indeed perform zero-shot separation given appropriate textual descriptions of the input during inference, it requires train a text-conditioned diffusion model on mixtures only, which is not as versatile as our method where any off-the-shelf text-guided diffusion model can work. Also, GMSDI's effectiveness heavily depends on the quality and relevance of the textual embeddings and the accuracy of the provided text descriptions. And the open-set ability is constrained due to the mixture dataset it train on. Moreover, as **no open-source implementation or pretrained checkpoints** for GMSDI are currently available, we were unable to include comparative results from this method in our evaluation during the rebuttal window. However, we will include a detailed discussion with GMSDI in the revised manuscript. We summarize the difference in the table below:
>
> | Aspect                 | GMSDI                                       | **ZeroSep (ours)**                                         |
> | ---------------------- | ------------------------------------------- | ---------------------------------------------------------- |
> | Mechanism              | Posterior *sampling* with Dirac likelihood  | *Latent inversion* followed by conditional denoising       |
> | Base model requirement | Text‑conditioned *mixture*‑only training    | Any pretrained T2A model (AudioLDM1, AudioLDM2, …) |
> | Conditioning interface | Needs *two* text prompts (mixture & target) | One prompt per target; no mixture text needed              |
> | Open‑set ability       | Bound by training mixture corpus            | Inherits full vocabulary of the base T2A model             |
>
>
> **Difference to AudioEditor.**
> AudioEditor targets *editing*; ZeroSep targets *separation*.  Beyond task difference, ZeroSep introduces **separation-based analysis** that focus on at what situations a T2A diffusion model can be repurposed to a separation model. These design principles are critical for accurate multi‑source separation; AudioEditor does not address them.
>
> **Insight on CFG Weighting.**
> We agree with the reviewer that generation-oriented CFG trade‑offs is well taken, where lower CFG reduce conditioning strength and higher CFG may degrade result quality. However we sweep CFG $\in$ [0, 2] and find CFG=1 yields good trade-off between separation and degradation (under-separation or over-generation, see Fig. 3(a)). This specific CFG choice is not well stuided as most papers focus on scenarios where CFG > 1. For example, Fig. 8 in (Karras et al. 2024) studies CFG=4 and Table 1 in (Postolache et al. 2024) stuides CFG=(3,7.5,15,24) We clarify this distinction and will add more discussion in the revised manuscript.
>
> ---
>
> ### Q3 Metric Suite Expansion
>
> **Citation of Jayaram et al. 2020**
> Thank the reviewer for providing the reference to better motivate our finding. We will cite Jayaram et al. 2020 in the Table 3.
>
> **Implementation of FAD**
> In our paper, we use the embedding space of CLAP model to compute FAD.
>
> **Improve Table 1 with More Metrics:**
> We thank the reviewer for the great suggestion! We agree that additional metrics like L1/L2 loss on Encodec embedding and MSE on mel-spectrogram can capture temporal fidelity better. Below we show the improved table:
>
> | Method | MUSIC FAD ↓ | MUSIC LPAPS ↓ | MUSIC C-A ↑ | MUSIC C-T ↑ | MUSIC MSE ↓ | MUSIC Encodec L1 (Emb.) ↓ | MUSIC Encodec L2 (Emb.) ↓ | AVE FAD ↓ | AVE LPAPS ↓ | AVE C-A ↑ | AVE C-T ↑ | AVE MSE ↓ | AVE Encodec L1 (Emb.) ↓ | AVE Encodec L2 (Emb.) ↓ |
> |---|---|---|---|---|---|---|---|---|---|---|---|---|---|---|
> | **Require Separation Training** | | | | | | | | | | | | | | |
> | LASS-Net | 1.039 | 5.602 | 0.204 | 0.014 | 32.860 | 0.6205 | 0.7338 | 0.626 | 6.062 | 0.232 | 0.011 | 12.612 | 0.5984 | 0.6737 |
> | AudioSep | 0.725 | 5.209 | 0.450 | 0.204 | **10.857** | 0.4232 | **0.4046** | 0.446 | 5.733 | 0.457 | 0.167 | 9.383 | **0.4949** | **0.5095** |
> | FlowSep | 0.402 | 5.578 | 0.564 | 0.245 | 15.134 | 0.5525 | 0.5869 | **0.258** | 4.719 | **0.493** | 0.082 | **7.785** | 0.6706 | 0.8402 |
> | **Separation Training Free** | | | | | | | | | | | | | | |
> | NMF-MFCC | 1.286 | 5.618 | 0.239 | -0.055 | 36.761 | **0.4230** | 0.4180 | 1.246 | 5.851 | 0.174 | **0.211** | 59.594 | 0.5766 | 0.6906 |
> | AudioEdit | 0.568 | 4.869 | 0.453 | 0.196 | 29.357 | 0.8059 | 1.3088 | 0.372 | 4.959 | 0.341 | 0.074 | 22.253 | 0.9588 | 1.7090 |
> | ZeroSep (Ours) | **0.377** | **4.669** | **0.615** | **0.271** | 15.470 | 0.5540 | 0.5912 | 0.269 | **4.537** | 0.442 | -0.001 | 11.076 | 0.7959 | 1.1258 |
>
> Generally, on the newly added metrics, ZeroSep matches the supervised method FlowSep and outperform LASS-net on the MUSIC dataset, confirming genuine separation rather than prompt‑consistent hallucination.
>
> ---
>
> ### Q5 Minor Corrections & Wording
> We will incorporate the following edits verbatim:
>
> | Line | Fix                                                                                                                                                                                |
> | ---- | ---------------------------------------------------------------------------------------------------------------------------------------------------------------------------------- |
> |  125 | Bold subscripts → normal font                                                                                                                                                      |
> |  127 | “parameterized by a set of weights \$\theta\$”                                                                                                                                     |
> |  128 | “reversing the diffusion **process**”                                                                                                                                              |
> |  128 | Clarify decoding step for AudioLDM2; note that Stable Audio Open operates in waveform space                                                                                        |
> |  155 | Specify it as our particular setting                                                                                                                                 |
> |  163 | Rephrase: “While separation has often been framed as a discriminative task, we follow the recent line of *generative* source separation methods.” |

---

> > ### Author Response · Authors · 2025-08-06
> >
> > Hi! We've attached our response here. Please don’t hesitate to reach out if you have any questions or suggestions :) Thank you so much for your time and effort in reviewing our paper and helping us improve it!!

---

> ### Comment · Reviewer_V2hC · 2025-08-06
>
> Dear authors, thank you for adressing my concerns: 1.  the novelty claim 2. adding missing references 3. adding L1 / L2 metrics on embeddings. While I am a bit skeptical on the L1 / L2 losses results (why NMF-MFCC gets such a good score on this being a relatively weak method?), I decided to raise the score to Borderline Accept.

---

> > ### Author Response · Authors · 2025-08-08
> >
> > Hi Reviewer V2hC,
> >
> > Thank you for improving the score to a positive rating! Regarding the L1/L2 results of NMF-MFCC, we have observed that NMF-MFCC often produces silence or heavily smoothed outputs, a failure mode distinct from other comparative methods, whose failures typically manifest as insufficient source separation rather than smoothed or empty results. In this context, NMF-MFCC can appear to “cheat” the Encodec embedding distance: its smoothed, low-detail outputs happen to align more closely with Encodec’s coarse embedding space, even though much of the fine-grained content is missing. This might explain why NMF-MFCC achieves seemingly strong results in the L1/L2 embedding metrics, yet performs poorly in other semantic and perceptual metrics such as FAD, LPAPS, and mel-spectrogram MSE.

---

### Official Review · Reviewer_KHgX · 2025-06-23

**Clarity:** 4
**Significance:** 3
**Originality:** 2
**Rating:** 4
**Confidence:** 4

**Summary:**

The paper introduces ZeroSep, a novel zero-training framework for audio source separation that repurposes pre-trained text-guided audio diffusion models. Unlike traditional supervised methods, ZeroSep leverages latent inversion and text-conditioned denoising to separate audio sources without task-specific training. The key insight is that a pre-trained diffusion model, originally designed for generation, can be steered via text prompts to disentangle mixed audio into individual sources. Experiments on AVE and MUSIC datasets show that ZeroSep outperforms supervised baselines, demonstrating its effectiveness in open-set scenarios. The framework offers three main advantages: open-set separation capability, model-agnostic versatility, and training-free efficacy.

**Questions:**

See weaknesses.

**Ethical Concerns:**

["NO or VERY MINOR ethics concerns only"]

**Final Justification:**

The author proposed an audio separation method that does not require training and has advantages in audio separation tasks. The drawback is that although the separation task is not the same as the extraction task in audio edit, it is still quite similar in itself.

**Limitations:**

yes

**Quality:**

3

**Strengths And Weaknesses:**

## Strengths
1. Clear Mechanistic Analysis: The paper thoroughly investigates the impact of key components (inversion strategies, guidance weight, prompt selection), providing insights into how generative models can be adapted for separation. The correlation between generative fidelity and separation performance is particularly noteworthy.
2. Open-Set Capability: By leveraging text-guided priors, ZeroSep naturally handles unseen source types, a critical advantage for real-world acoustic scenes. The framework’s compatibility with different diffusion backbones (e.g., AudioLDM, Tango) showcases its versatility.
3. Conceptual Innovation: ZeroSep represents a paradigm shift by repurposing generative diffusion models for discriminative separation tasks, eliminating the need for supervised training. This approach addresses the data scarcity and generalization challenges of traditional methods.

## Weaknesses
1. Latent Inversion Limitations: The inversion process may introduce approximation errors, as acknowledged by the authors. While both DDIM and DDPM inversion are tested, the impact of these errors on separation fidelity—especially for complex mixtures—requires deeper analysis.
2. Prompt Engineering Dependence: Effective separation relies on precise text prompts. The paper demonstrates performance with ideal prompts but does not address how suboptimal or ambiguous prompts affect results, nor does it explore automated prompt generation in depth.
3. Lack of novelty: The author emphasizes that the model is used for audio separation, so the baseline systems compared in the experimental and method sections are very limited. In fact, separation is a subtask of audio editing, and authors should compare more editing models to demonstrate the innovation and effectiveness of their methods. Because there are many very similar methods in the field of audio editing[1][2][3][4], the author should **analyze and emphasize differentiation** between the proposed method and these methods, especially for [1][2].

[1] Meng C, He Y, Song Y, et al. Sdedit: Guided image synthesis and editing with stochastic differential equations[J]. arXiv preprint arXiv:2108.01073, 2021.

[2] Jia Y, Chen Y, Zhao J, et al. AudioEditor: A Training-Free Diffusion-Based Audio Editing Framework[J]. arXiv preprint arXiv:2409.12466, 2024.

[3] Liu H, Wang J, Li X, et al. MEDIC: Zero-shot Music Editing with Disentangled Inversion Control[J]. arXiv preprint arXiv:2407.13220, 2024.

[4] Zhang Y, Ikemiya Y, Xia G, et al. MusicMagus: zero-shot text-to-music editing via diffusion models[C]//Proceedings of the Thirty-Third International Joint Conference on Artificial Intelligence. 2024: 7805-7813.

---

> ### Author Rebuttal · Authors · 2025-07-30
>
> **We thank the reviewer for highlighting our mechanistic analysis, open‑set capability, and the “paradigm‑shift” use of text‑guided diffusion for separation.** We clarify and address each question below.
>
> ### Q1. Latent inversion approximation and mitigation
>
> DDIM inversion is fast but introduces **stepwise drift**. As the reviewer mentioned, MEDIC found DDIM inversion works for very small edits, yet it accumulates reconstruction error, breaks musical structure, and offers no fine‑grained control. MEDIC tackles all three pain points with a plug‑and‑play framework that needs no extra training. Therefore, MEDIC pointed out a direction to mitigate the latent inversion error, which will also apply to our method. Apart from MEDIC, there are more correction methods that exist in vision. ZeroSep is drop‑in compatible with each:
>
> * **Null‑text (embedding) optimization**: optimizes the unconditional embedding along the trajectory to match the source mixture, compensating for score error.
> * **PnP‑style inversions**: explicitly estimate and inject residual noise to correct drift.
> * **DDPM‑style stochastic inversion**: by recovering per‑step noise latents, it better matches the forward process and reduces accumulation compared with purely deterministic DDIM inversion.
>
> We currently use a DDPM‑based inversion variant precisely to **mitigate drift**; we will expand the discussion on inversion methods and possible mitigation.
>
> ### Q2.  Prompt sensitivity & hierarchical/open‑set behavior
>
> We probe 11 instruments with two perturbations per class:
>
> * **Ambiguous (hypernym / family):** *violin → string instrument*, *flute → woodwind*.
> * **Suboptimal (underspecified / distractor adj.):** *trumpet → bright instrument*, *cello → low tone*.
>
> **Results (vs. canonical prompts).**
>
> | prompts    |              FAD ↓ | LPAPS ↓ |              C‑A ↑ |              C‑T ↑ |
> | ---------- | -----------------: | ------: | -----------------: | -----------------: |
> | canonical  |              0.377 |   4.669 |              0.615 |              0.271 |
> | ambiguous  | 0.388 (**+0.011**) |   4.692 | 0.583 (**−0.032**) | 0.280 (**+0.009**) |
> | suboptimal | **0.336 (−0.041)** |   4.714 |     0.577 (−0.038) |     0.248 (−0.023) |
>
> **Interpretation.** Hypernyms keep the correct **family**, yielding small degradations; the slight **C‑T rise** suggests the classifier judges family‑level matches as more consistent, while **C‑A** (accuracy to ground‑truth class) drops. Suboptimal descriptors sometimes improve **distributional proximity (FAD)** yet hurt **class accuracy**, indicating plausible but less precise extractions. We will add this table and clarify metric roles.
>
> ### Q3. Novelty vs. editing‑oriented methods \[1‑4]**
>
> | Method             | Task                 | Per‑clip optimization  | Separation analysis                  |
> | ------------------ | -------------------- |  -------------- | ---------------------------------- |
> | SDEdit \[1]        | **Image** editing    | ✗              | ✗                                 |
> | AudioEditor \[2]   | Audio **editing**    | **✓ (> 1 min)** | ✗                        |
> | MEDIC \[3]         | Music editing        | ✗              | ✗                                  |
> | MusicMagus \[4]    | Music editing        | ✗              | ✗                                 |
> | **ZeroSep (Ours)** | **Audio separation** | ✗              | **✓**  |
>
> Key differences:
>
> 1. **Task** – We target *source separation*, not waveform replacement. Separation demands preserving mixture consistency and yielding *N* disjoint sources, which editing pipelines cannot guarantee (see Table 1 AudioEdit row)
> 2. **Training‑free *and* optimization‑free** – Unlike AudioEditor’s per‑prompt null‑text fitting, ZeroSep is runtime‑constant.
> 3. **Separation analysis** – Our paper dissects the role of inversion, guidance weight, and prompt specificity—analysis particularly related to separation task, yielding different findings compared to \[1‑4].
>
> Nonetheless, we will add \[1‑4] to Related Work to include a in-depth discussion.

---

> > ### Comment · Reviewer_KHgX · 2025-08-06
> >
> > Thank you for your reply, it has to some extent answered my question.

---

> > > ### Author Response · Authors · 2025-08-06
> > >
> > > Thank you for your follow-up and for taking the time to engage with our rebuttal!! We're glad to hear that some of your questions were addressed, and we’d be happy to clarify or expand on any remaining concerns if helpful :)

---

### Official Review · Reviewer_2kNp · 2025-07-01

**Clarity:** 2
**Significance:** 2
**Originality:** 3
**Rating:** 4
**Confidence:** 5

**Summary:**

### **Summary**

This paper introduces a novel audio source separation framework named ZeroSep, whose core idea is to leverage a pre-trained text-guided audio diffusion model to achieve the separation of arbitrary audio sources without any additional training or fine-tuning. The main mechanism of this method consists of two steps: first, through a diffusion inversion process, the mixed audio to be separated is mapped into the latent noise space of the diffusion model; then, a text prompt describing the target source is used to guide the denoising process, thereby reconstructing a single target source from the noise.

**Questions:**

### **Questions**

1.  Are the datasets used for evaluation too few or lacking in generality? Most research on general-purpose separation uses AudioSet for training and evaluation, including the selected baselines LASSNet, AudioSep, and Flowsep. How does ZeroSep perform on this dataset? Does it still outperform traditional methods? Furthermore, the original papers for all the baselines did not use the AVE dataset. Is the choice of this dataset somewhat niche or less conventional?

2.  Additionally, since prompts are used, is it possible to use descriptive captions for separation instead of just simple class labels? Does the length or complexity of the prompt content affect the framework's performance?

3.  Regarding the separation demo shown in the paper (Figure 2), is ZeroSep's performance really significantly superior to the other methods in this specific example? Could a more compelling example have been chosen? Also, it appears that the spectrogram for the LASSNet result is displayed upside down.

**Ethical Concerns:**

["NO or VERY MINOR ethics concerns only"]

**Final Justification:**

My final score is 4 (weakly accepted), and my concerns have been addressed.

**Limitations:**

yes

**Quality:**

3

**Strengths And Weaknesses:**

### **Strengths**

1.  It directly applies a pre-trained generative diffusion model to a discriminative separation task without any training, which is a departure from the current mainstream supervised separation paradigm that relies on large amounts of paired data.
2.  It explores the inconsistent role of the classifier-free guidance weight in the separation task compared to its role in generative tasks.
3.  ZeroSep's performance on multiple benchmark datasets is comparable to or even surpasses supervised separation models, providing a new paradigm for open-set audio separation.

### **Weaknesses**

1.  Diffusion models typically have high computational costs during inference, as they require a multi-step iterative denoising process. The ZeroSep framework needs to perform a complete 'inversion + denoising' process for *each* target source in the mixture. This means that separating a mixture containing N sources requires approximately N times the computation of generating a single audio clip. The paper does not mention or compare the inference time differences between ZeroSep and the baseline methods at all.

2.  The experiments show that using a null prompt (cinv = 'none') performs better than using a prompt that describes other (non-target) sources in the mixture. This seems somewhat counter-intuitive, as providing prior information about the background sounds should theoretically help the model perform a better inversion.

3.  In the ablation study for `crev`, the conclusion does not fully align with the experimental results. On the AVE dataset, the CLAP-T metric improved when using a random prompt, so the claim that performance dropped "across all metrics" is inaccurate.

4.  The ablation study for `cinv` does not seem to lead to a clear conclusion; it feels like it was just performed for the sake of it. Theoretically, changing `cinv` to the label of a non-target audio in the mixture should improve performance, but the actual results do not support this, and this discrepancy was not further analyzed.

---

> ### Author Rebuttal · Authors · 2025-07-30
>
> **We thank the reviewer for highlighting ZeroSep’s novelty and for the constructive questions.** We clarify and address each point below.
>
> ### Q1. Inference cost
>
> Let $S$ be the number of network inference steps and $T_d$ the time per step. ZeroSep performs **one inversion** of the mixture (shared for all separation branches) and then runs one guided denoising per target source. Thus complexity is
>
> $$
> T_{\text{ZeroSep}} \approx \underbrace{S\,T_d}_{\text{inversion}}+\underbrace{N\,S\,T_d}_{\text{N targets}}=(N+1)S\,T_d.
> $$
>
> For single‑pass discriminative models like LASS-Net and AudioSep, the runtime is ${N\,T_d}$ which only 1 step inference is required. For FlowSep, which also based on a rectified flow model and requires multi-step inference, its runtime is $N\,S\,T_d$ for multi‑target separation.
> In summary, we list the table below with the runtime complexiity for each method and the actual time cost on a A100 GPU.
>
> | Method                            | Complexity  | Avg. Runtime (Separate one source on A100)                         |
> | --------------------------------- | ----------- | ---------------------------- |
> | LASS‑Net                          | ${N\,T_d}$        | 0.05 s                   |
> | Audio-Sep                         | ${N\,T_d}$        | 0.04 s                   |
> | FlowSep                           | ${N\,S\,T_d}$      | 1.07 s                       |
> | ZeroSep (AudioLDM-S)              | ${N\,S\,T_d}$  | 0.98 s                       |
>
>
> Note that $S$ and $T_d$ for different methods is different. In our implementation, ZeroSep’s runtime is larger than LASS-Net and Audio-Sep while keeping similar as FlowSep. In practice, **$S$** can be reduced substantially with advanced samplers or distillation: **DPM‑Solver++** attains high quality in **15–20 steps**, and **consistency models** reduce to **2–8 or even 1 step**, which directly scales our runtime down.
>
> ### Q2/Q4. Why a null inversion prompt outperforms non‑target prompts
>
> A non‑target prompt biases the latent inversion toward irrelevant sources, constricting the representation available for the true target during guided denoising. A null prompt leverages the diffusion prior’s rich, unbiased mixture representation, after which the target‑specific prompt can act more effectively. Table 4 shows a consistent – albeit modest – drop when conditioning on non‑targets. We will make this intuition explicit.
>
> ### Q3.  $c_{\text{rev}}$ ablation wording
> Thank you for point out this issue. Indeed, the CLAP‑T score on AVE improves with a random prompt. We will replace “across all metrics” with “across all metrics except CLAP‑T on AVE”
>
>
> ### Q5. Dataset choice
> We clarify that our dataset choice is reasonable:
> * First, the two datasets, MUSIC and AVE, are the de‑facto multimodal (including text query) source separation benchmarks [1, 2, 3, 4]. Therefore, they are suitable for assessing the performance of unsupervised or text-guided separation methods as shown in the Tab 1.
>
> [1] The Sound of Pixels
> [2] CLIPSep: Learning Text-queried Sound Separation with Noisy Unlabeled Videos
> [3] iQuery: Instruments As Queries for Audio-Visual Sound Separation
> [4] Language-Guided Audio-Visual Source Separation via Trimodal Consistency
>
> * Second, not evaluating on AudioSet can test open‑set generalization since all supervised baselines except FlowSep were trained on AudioSet yet are evaluated out‑of‑domain here. ZeroSep’s gains therefore reflect true open domain robustness.
>
> ### Q6 │ Prompt length / descriptiveness
>
> We ran a focused prompt‑descriptiveness study on the MUSIC dataset. Specifically, We expanded each of the 11 MUSIC instrument labels into a free‑form descriptive caption (~10x length). For example: "saxophone" to "Warm, breathy reed tone with slight rasp and expressive bends,"accordion" to "Reedy, wheezy sustained chords with gentle tremolo and slow swells." This create a challenging setting to test ZeroSep where the prompt is descriptive but potentially noisy.
>
> | Prompt type | FAD ↓     | LPAPS ↓   | CLAP‑A ↑  | CLAP‑T ↑  |
> | ----------- | --------- | --------- | --------- | --------- |
> | Class label | **0.377** | 4.669     | **0.615** | **0.271** |
> | Descriptive | 0.484     | **4.541** | 0.512     | 0.196     |
> | AudioSep    | 0.725     | 5.209     | 0.450     | 0.204     |
>
> We observe a moderate drop of ZeroSep yet still beat the strong supervised baseline AudioSep. We agrue that the degradtation likely stems from a mismatch between our expansion and the caption used to train Text-to-Audio diffusion model, which is often a simple audio description like "a man speak while the baby is crying".
>
> ### Q7. Qualitative demo (Fig. 2)
> Thank you for suggesting improvement on the choice of qualitative example. We have included additional qulitative examples in the supplement, and we will correct LASSNet’s spectrogram orientation in the revised manuscript plus adding more qualitative examples in the main paper.

---

> > ### Author Response · Authors · 2025-08-06
> >
> > Hi! We've attached our response here. Please don’t hesitate to reach out if you have any questions or suggestions :) Thank you so much for your time and effort in reviewing our paper and helping us improve it!!

---

> > > ### Comment · Reviewer_2kNp · 2025-08-08
> > >
> > > I appreciate the authors' response. May I ask if higher-order samplers, such as DPM-Solver, or distillation methods, such as consistency models, could be used for inference acceleration?

---

> > > > ### Author Response · Authors · 2025-08-08
> > > >
> > > > Thank you for the follow-up question — it’s a good point! In our paper, we support two inversion methods: **DDIM** and **DDPM inversion**. DDPM inversion has the property of *perfect inversion*, which means that inference acceleration techniques such as **DPM-Solver++** can be directly integrated into our framework. This has been demonstrated in prior work using DDPM inversion for image editing \[1], and in our future code release, we have already incorporated **DPM-Solver** acceleration.
> > > >
> > > > As for consistency models, their integration into our framework remains an open direction. In principle, they could further reduce inference to just a few steps (or even a single step), but we have yet to empirically validate their effectiveness for our separation setting, something we plan to explore in future work.
> > > >
> > > > \[1] *Ledits++: Limitless Image Editing using Text-to-Image Models*, CVPR 2024.

---

### Official Review · Reviewer_rKJV · 2025-07-02

**Clarity:** 4
**Significance:** 4
**Originality:** 3
**Rating:** 5
**Confidence:** 4

**Summary:**

Authors propose the first latent inversion approach to sound-source separation, and by virtue of a completely training-free approach. Particularly, ZeroSep repurposes pretrained generative diffusion models for discriminative separation tasks without any task-specific training, representing a fundamental departure from past supervised/finetuning approaches.

**Questions:**

Some minimal exploration on optimal text conditioning for different types of acoustic scenes, and how sensitive the method is to prompt phrasing, would help improve the quality of this paper.

Can authors comment on the hierarchical boundaries of events? Specifically, what is the permissible deviation of text prompts from the training taxonomy while still ensuring effective separation performance? It would be interesting to see the extent of open-set separation, say, distinguishing unseen leaf node events that belong to already seen parent events. Open-set separation is considered a particularly strong claim.

**Ethical Concerns:**

["NO or VERY MINOR ethics concerns only"]

**Limitations:**

yes.

**Paper Formatting Concerns:**

no.

**Quality:**

3

**Strengths And Weaknesses:**

Strengths
ZeroSep surpasses existing training-based generative methods on multiple benchmarks - all metrics on MUSIC dataset, and comparable to flowsep on AVE -while being training-free.
More importantly, the overall approach is model-agnostic and is bound to improve with better T2A models in the future.
Overall, paper is well-written with relevant references and easy to read-understand.

Weakeness
While respecting the page limit, the authors fail to touch on some theoretical analysis explaining why T2A diffusion models possess inherent separation capabilities; Is this an emergent property of the generative objective? -Some analysis regarding how much percentage of T2A model’s training data included polyphonic sounds would be helpful.

Similarly, \omega = 1 removes unconditional noise estimation deserves more rigorous theoretical grounding; model follow only the conditional score? Is it the case that higher \omega pushes off the manifold? Do you think a dynamic guidance schedule adds further to improving performance? One quick suggestion would be to test a simple linear or cosine ramp that peaks at 1 around the middle 40-60% of steps.

Can the authors provide detailed runtime comparisons + scalability analysis for separating multiple sources simultaneously compared to single-pass (non-diffusion) methods?

The paper mentions that latent inversion can introduce approximation errors that may impair separation fidelity, but it does not thoroughly investigate how these errors accumulate or propose mitigation strategies. MEDIC argues that imprecise approximation of the original audio using DDIM inversion can allow authors to comment on the same, or argue why ZeroSep is not impacted by the assumption of perfect reversibility of ODE, often not met during text-based conditioning?

Please provide analysis on complex polyphonic mixtures involving more than 2 sources/instances of the same event class.

---

> ### Author Rebuttal · Authors · 2025-07-30
>
> **We thank the reviewer for acknowledging the novelty and importance of our paper, and also for providing constructive comments to help improve the paper.** Below, we address each point.
>
> ### Q1. Why do T2A diffusion models exhibit zero‑shot separation ability?
>
> **Data perspective.** Text‑to‑audio (T2A) diffusion models are usually trained on **multi‑label, polyphonic** audio corpora, so the model must learn *both* the acoustics of individual sources and how they combine. Thus, the model learns *composable* latent factors that align with text semantics (e.g., *dog bark*, *rain*) and their superposition (e.g., *a dog bark and raindrops*). This polyphonic nature exists in many T2A pretraining datasets. For example, AudioSet -- the predominant pretraining resource -- has **2.7 labels per 10‑s clip on average**[1], indicating widespread mixtures, which encourages the model to encode disentangled source structure.
>
> [1] Audio set: An ontology and human-labeled dataset for audio events
>
> **Score-function view.** Classifier‑free guidance (CFG) combines an unconditional score $\nabla_{x_t}\log p(x_t)$ with a conditional increment that steers toward $\nabla_{x_t}\log p(c\mid x_t)$:
>
> $$
> \hat\epsilon_\theta(x_t,c)=\epsilon_\theta(x_t,\varnothing)
> + \omega\big(\epsilon_\theta(x_t,c)-\epsilon_\theta(x_t,\varnothing)\big),
> $$
>
> which is a standard derivation in CFG. Setting $\omega=1$ removes the unconditional term and retains only the conditional direction, effectively **filtering the inverted latent toward the target source** specified by text. Therefore, the zero-shot separation capability is an emerging property of T2A diffusion models trained with mixture audio. We will add this discussion to the revised manuscript.
>
> ### Q2. Effect of $\omega$ and scheduled guidance
>
> Large $\omega$ amplifies the conditional score. Because both scores are learned approximations, excessive scaling can move samples toward low‑probability regions (“off the data manifold”), improving prompt adherence and diversity of generated samples but risking artifacts -- an effect documented broadly in image/audio diffusion [2, 3].Therefore, it will opt for generation instead of accurate separation with large $\omega$. This potential artifact, along with large $\omega$, does not contribute to accurate separation, and empirically we show $\omega=1$ achieves a good trade-off between removing the other sounds and introducing generated content/artifact.
>
> **Static vs. scheduled guidance.** Thank you for the great suggestion! Intuitively, the early stage of diffusion models builds the global structure, while the later denoising steps act as a refinement stage. We tested constant and scheduled $\omega$ on **AudioLDM2‑Large** backbone (DDIM, 50 steps). Results on the MUSIC dataset:
>
> | ω schedule |     FAD ↓ |   LPAPS ↓ |     C‑A ↑ |     C‑T ↑ |
> | ---------- | --------: | --------: | --------: | --------: |
> | constant 1 |     0.377 |     4.669 |     0.615 |     0.271 |
> | linear 0→1 |     0.471 | **4.526** |     0.520 |     0.202 |
> | linear 1→0 | **0.332** |     4.667 | **0.618** | **0.281** |
> | sine 0→1 |     0.431 |     4.523 |     0.550 |     0.227 |
>
>
> **Observations.** (i) **Low-to-high**, Starting at 0 (0→1, sine) harms separation (global structure lost of clean target audio). (ii) **High‑to‑low (1→0)** improves over constant 1.0, aligning with findings that guidance is most useful in the **early to middle** noise range and less beneficial at the end. This further enhances our claim that $\omega=1$ is beneficial for accurate separation, while dynamic scheduling based on this finding can also be promising in future exploration, which we will add discussion in the revision.
>
> [2]: CLASSIFIER-FREE DIFFUSION GUIDANCE. NeurIPS 2021 Workshop
>
> [3]: ELIMINATING OVERSATURATION AND ARTIFACTS OF HIGH GUIDANCE SCALES IN DIFFUSION MODELS. ICLR 2025.
>
> ### Q3. Runtime and scalability for multiple sources
>
> Let $S$ be the number of network inference steps and $T_d$ the time per step. ZeroSep performs **one inversion** of the mixture (shared for all separation branches) and then runs one guided denoising per target source. Thus complexity is
>
> $$
> T_{\text{ZeroSep}} \approx \underbrace{S\,T_d}_{\text{inversion}}+\underbrace{N\,S\,T_d}_{\text{N targets}}=(N+1)S\,T_d.
> $$
>
> For single‑pass discriminative models like LASS-Net and AudioSep, the runtime is ${N\,T_d}$ where only 1 step inference is required. For FlowSep, which is also based on a rectified flow model and requires multi-step inference, its runtime is $N\,S\,T_d$ for multi‑target separation.
> In summary, we list the table below with the runtime complexity for each method and the actual time cost on an A100 GPU.
>
> | Method                            | Complexity  | Avg. Runtime (Separate one source on A100)                         |
> | --------------------------------- | ----------- | ---------------------------- |
> | LASS‑Net                          | ${N\,T_d}$        | 0.05 s                   |
> | Audio-Sep                         | ${N\,T_d}$        | 0.04 s                   |
> | FlowSep                           | ${N\,S\,T_d}$      | 1.07 s                       |
> | ZeroSep (AudioLDM-S)              | ${N\,S\,T_d}$  | 0.98 s                       |
>
>
> Note that $S$ and $T_d$ for different methods are different. In our implementation, ZeroSep’s runtime is larger than LASS-Net and Audio-Sep while remaining similar as FlowSep. In practice, **$S$** can be reduced substantially with advanced samplers or distillation: **DPM‑Solver++** attains high quality in **15–20 steps**, and **consistency models** reduce to **2–8 or even 1 step**, which directly scales our runtime down.
>
>
>
>
> ### Q4. Latent inversion approximation and mitigation
>
> DDIM inversion is fast but introduces **stepwise drift**. As the reviewer mentioned, MEDIC found DDIM inversion works for very small edits, yet it accumulates reconstruction error, breaks musical structure, and offers no fine‑grained control. MEDIC tackles all three pain points with a plug‑and‑play framework that needs no extra training. Therefore, MEDIC pointed out a direction to mitigate the latent inversion error, which will also apply to our method. Apart from MEDIC, there are more correction methods that exist in vision. ZeroSep is drop‑in compatible with each:
>
> * **Null‑text (embedding) optimization**: optimizes the unconditional embedding along the trajectory to match the source mixture, compensating for score error.
> * **PnP‑style inversions**: explicitly estimate and inject residual noise to correct drift.
> * **DDPM‑style stochastic inversion**: By recovering per‑step noise latents, it better matches the forward process and reduces accumulation compared with purely deterministic DDIM inversion.
>
> We currently use a DDPM‑based inversion variant precisely to **mitigate drift**; we will expand the discussion on inversion methods and possible mitigation.
>
> ### Q5. Polyphonic mixtures
>
> ZeroSep supports **multi‑source separation** naturally. We additionally include results on **3-source** mixtures built from MUSIC dataset (e.g., violin+flute+trumpet), which shows moderate degradation as the number of mixing sources grows.
>
> | # sources | FAD ↓ | LPAPS ↓ | C‑A ↑ | C‑T ↑ |
> | :-------: | ----: | ------: | ----: | ----: |
> |     2     | 0.377 |   4.669 | 0.615 | 0.271 |
> |     3     | 0.508 |   4.248 | 0.501 | 0.246 |
>
> For the challenging cases the reviewer mentions, where sources from the same event class are mixed, we argue that adding **attribute qualifiers** in text (e.g., “*soft violin arpeggio*” vs. “*long sustained violin note*”) could help. We will include the new 3-source separation results and add the discussion on challenging instance-level separation in the revised manuscript.
>
> ## Q6. Prompt sensitivity & hierarchical/open‑set behavior
>
> To test the prompt sensitivity and hierarchical prompting performance, given the class labels from the MUSIC dataset, we probe 11 instrument class names with two perturbations per class:
>
> * hypernyms:** *violin → string instrument*, *flute → woodwind*.
> * underspecified descriptors:** *trumpet → bright instrument*, *cello → low tone*.
>
> **Results (vs. canonical prompts).**
>
> | Prompt type    |              FAD ↓ |        LPAPS ↓ |          C‑A ↑ |              C‑T ↑ |
> | -------------- | -----------------: | -------------: | -------------: | -----------------: |
> | canonical      |              0.377 |          4.669 |          0.615 |              0.271 |
> | hypernym       |     0.388 (+0.011) | 4.692 (+0.023) | 0.583 (−0.032) | **0.280 (+0.009)** |
> | underspecified | **0.336 (−0.041)** | 4.714 (+0.045) | 0.577 (−0.038) |     0.248 (−0.023) |
>
>
> **Observation.** Hypernyms keep the correct **family**, yielding small degradations; the slight **C‑T rise** suggests the classifier judges family‑level matches as more consistent, while **C‑A** (accuracy to ground‑truth class) drops. Underspecified descriptors sometimes improve **distributional proximity (FAD)** yet hurt **class accuracy**, indicating plausible but less precise extractions. We will add this table and clarify metric roles.
>
> **Open‑set note.** These results indicate **good generalization** from parent nodes (e.g., *string instrument*) to leaves (e.g., a violin articulation), but we avoid over‑claiming “strong open‑set separation.” We will frame it as **hierarchical robustness** and include success/failure cases in the revised manuscript.

---

> > ### Author Response · Authors · 2025-08-06
> >
> > Hi! We've attached our response here. Please don’t hesitate to reach out if you have any questions or suggestions :) Thank you so much for your time and effort in reviewing our paper and helping us improve it!!

---

### Decision · Program_Chairs · 2025-09-17

**Decision:**

Accept (poster)

**Comment:**

Summary:

The paper addresses the issue of open set source separation by leveraging a pre-trained text-to-audio diffusion model. The method uses text conditioning to guide the denoising process in the diffusion latent space (latent inversion approach to sound/source separation). The method is also model-agnostic, with the potential to improve with better generative models.

Strengths:

- Promising new paradigm for open set audio separation.
- Good results when compared with existing training-based generative methods.
- Explores the role of classifier-free guidance (CFG) in the context of sound/source separation, which is not the standard understanding of CFG. Also interesting is the combination of inversion strategies with guidance weight and prompt selection.
- Well-written paper and relevant references (after reviewers' suggestions).

Weaknesses:

- Lack of intuition/motivation on why text-to-audio generative models should possess some sort of source separation capabilities. Of course the results provide part of the justification, but one would like to understand further...
- Computational cost (this is extensively addressed by the authors in the rebuttal).
- Possible prompt engineering dependence.

Reasons to accept/reject:

I think the novelty and the new paradigm for open set audio separation is already a very good reason to accept the paper. The authors also provide convincing results.

Discussion and rebuttal:

The authors mostly addressed all reviewers' comments and concerns. They also provided some partial justification on the lack of intuition/motivation in the rebuttal, which they promise to add to the main paper. In general, they provided very detailed and informed answers (and tables/results), which I hope they can add to the main paper, perhaps removing some other less important parts.